# CAT-3DGS: A Context-Adaptive Triplane Approach to Rate-Distortion-Optimized 3DGS Compression

**Yu-Ting Zhan*[1], Cheng-Yuan Ho*[1], Hebi Yang[1], Yi-Hsin Chen[1], Jui Chiu Chiang[2]
Yu-Lun Liu[1], Wen-Hsiao Peng[1]**
[1]National Yang Ming Chiao Tung University, Taiwan
[2]National Chung Cheng University, Taiwan
{dotori25.ii12, kelvinhe0218.cs12}@nycu.edu.tw
{mrrrimge32.cs13, yhchen12101.cs09}@nycu.edu.tw
rachel@ccu.edu.tw
{yulunliu,wpeng}@cs.nycu.edu.tw
* Contributed equally

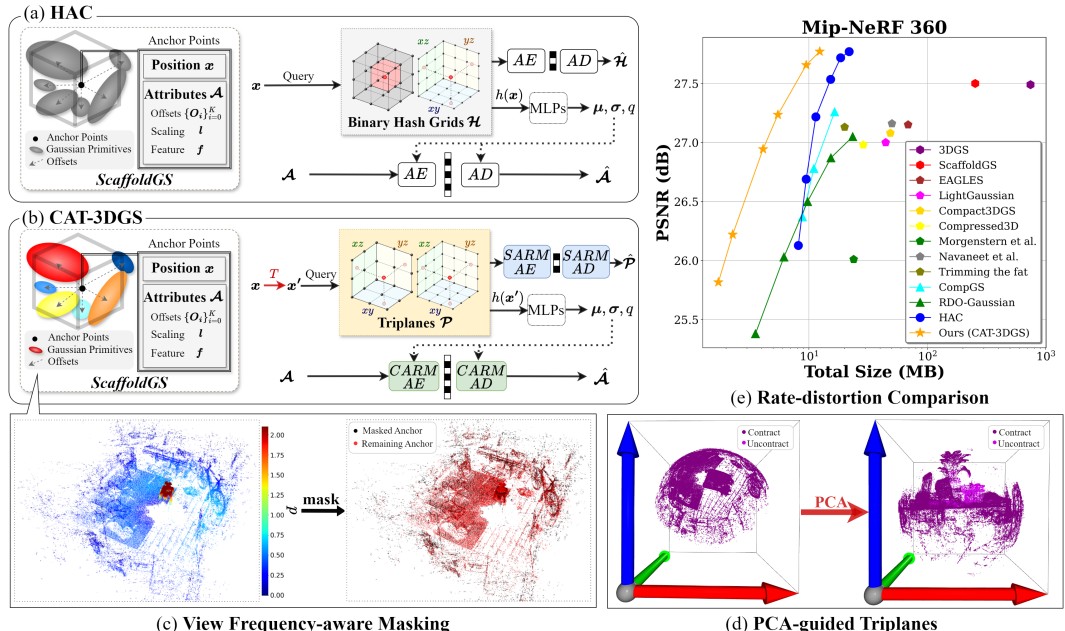

Figure 1: Comparison of the proposed CAT-3DGS and HAC (Chen et al., 2024). CARM: Channel-wise Autoregressive Models. SARM: Spatial Autoregressive Models.

## ABSTRACT

3D Gaussian Splatting (3DGS) has recently emerged as a promising 3D representation. Much research has been focused on reducing its storage requirements and memory footprint. However, the needs to compress and transmit the 3DGS representation to the remote side are overlooked. This new application calls for rate-distortion-optimized 3DGS compression. How to quantize and entropy encode sparse Gaussian primitives in the 3D space remains largely unexplored. Few early attempts resort to the hyperprior framework from learned image compression. But, they fail to utilize fully the inter and intra correlation inherent in Gaussian primitives. Built on ScaffoldGS, this work, termed CAT-3DGS, introduces a context-adaptive triplane approach to their rate-distortion-optimized coding. It features multi-scale triplanes, oriented according to the principal axes of Gaussian primitives in the 3D space, to capture their inter correlation (i.e. spatial correlation) for spatial autoregressive coding in the projected 2D planes. With these triplanes serving as the hyperprior, we further perform channel-wise autoregressive coding to leverage the intra correlation within each individual Gaussian prim-

itive. Our CAT-3DGS incorporates a view frequency-aware masking mechanism. It actively skips from coding those Gaussian primitives that potentially have little impact on the rendering quality. When trained end-to-end to strike a good rate-distortion trade-off, our CAT-3DGS achieves the state-of-the-art compression performance on the commonly used real-world datasets.

# 1 INTRODUCTION

3D Gaussian Splatting (3DGS) (Kerbl et al., 2023) has emerged as a promising representation for 3D scenes. It lends itself to novel view synthesis particularly within differentiable rendering frameworks. Unlike Neural Radiance Fields (NeRF) (Mildenhall et al., 2021), which require many sampling points per pixel for volumetric rendering, 3DGS, as a rasterization-based method, uses 3D Gaussians as geometric primitives, achieving greater efficiency for real-time rendering and the state-of-the-art rendering quality.

Despite these advantages, the redundancy inherent in the 3DGS representation has prompted new research directions. Many prior works have been focused on a compact representation of Gaussian primitives. This category of methods aim at minimizing the parameter count or quantizing parameters to save storage space and memory footprint. These techniques include pruning insignificant Gaussian primitives (Lee et al., 2024; Fan et al., 2023; Girish et al., 2024; Ali et al., 2024; Wang et al., 2024a), vector quantizing their attributes (Lee et al., 2024; Fan et al., 2023; Navaneet et al., 2023; Niedermayr et al., 2024; Wang et al., 2024a; Morgenstern et al., 2023), developing compact latent representations for attributes (Girish et al., 2024), and representing sparse Gaussian primitives in a more structural way (Lu et al., 2024; Ren et al., 2024; Sun et al., 2024). However, most of them overlook the needs to transmit the compressed 3DGS representation to the remote side, which calls for entropy coding and rate-distortion-optimized compression.

Recently, the rate-distortion-optimized compression for 3DGS started to attract attention. Unlike the compact 3DGS representation, this new school of thought (Wang et al., 2024a; Liu et al., 2024; Chen et al., 2024; Wang et al., 2024b) aims to strike an optimized trade-off between the compressed bit rate and rendering image quality in an end-to-end manner. Built on the vanilla 3DGS representation, RDO-Gaussian (Wang et al., 2024a) adopts the entropy-constrained vector quantization to quantize the attributes (e.g. opacity, scales, rotations and colors) of each Gaussian primitive. Instead of performing vector quantization, HAC (Chen et al., 2024) and ContextGS (Wang et al., 2024b) turn to the ScaffoldGS (Lu et al., 2024) representation to perform scalar quantization with respect to the latent features of these attributes, a technique analogous to the well-established transform coding plus scalar quantization for image/video compression. To entropy encode the quantized features, both introduce the hyperprior from learned image compression (Ballé et al., 2018) to model their coding probabilities. In formulating the hyperprior, HAC (Chen et al., 2024) draws inspiration from BiRF (Shin & Park, 2024) to create multi-scale binary hash grids (Figure 1 (a)), whereas ContextGS learns a separate feature as the hyperprior for each individual Gaussian primitive. Both assume the components of the hyperpior are independent and identically distributed, in coding the hyperprior. Notably, ContextGS organizes Gaussian primitives in the 3D space in a hierarchical manner in order to benefit from the contextual coding of the quantized latent features. In this regard, CompGS (Liu et al., 2024) shares parallels with ContextGS.

This work introduces a novel rate-distortion-optimized compression framework for 3DGS (Figure 1 (b)). First, motivated by the tensor decomposition (Fridovich-Keil et al., 2023), we project the unorganized Gaussian primitives in the 3D space onto a set of multi-scale triplanes. These triplanes, oriented according to the principal components of the Gaussian primitives (Figure 1 (d)), serve as the hyperprior for coding their attributes in the latent space. Because they capture largely the *inter correlation* (i.e. spatial correlation) between the Gaussian primitives in the 3D space, we are able to encode efficiently the triplane-based hyperprior by spatial autoregressive models. This design aspect differs significantly from most existing techniques, in which the hyperprior is assumed to be factorial. Second, given that our triplane-based hyperprior has exploited much of the inter correlation between the Gaussian primitives, we decouple their coding dependency and encode their latent features independently by channel-wise autoregressive models. This avoids the challenge of having to organize sparse Gaussian primitives in the 3D space to leverage their inter correlation. Moreover, the *intra correlation* within each individual Gaussian primitive is explored for coding. Lastly, we

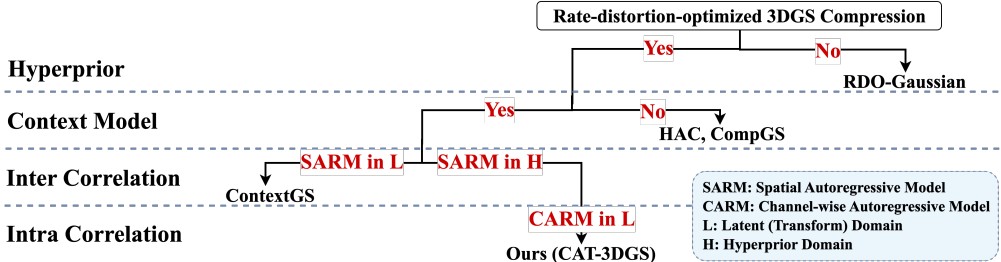

Figure 2: Taxonomy of the rate-distortion-optimized 3DGS compression.

develop a view frequency-aware masking mechanism, skipping from coding the Gaussian primitives that contribute little to the rendering quality (Figure 1 (c)). To sum up, our contributions include:

- A triplane-based hyperprior that leverages the inter correlation (i.e. spatial correlation) between Gaussian primitives in the 3D space for efficient spatial autoregressive coding.
- A channel-wise autoregressive model with uneven slice partition that exploits the intra correlation within each individual Gaussian primitive to further improve coding efficiency.
- A view frequency-aware masking mechanism that evaluates the significance of Gaussian primitives based on their impact on the rendering quality to skip less critical ones from coding.

With these novel elements, our scheme, called CAT-3DGS, is able to achieve the state-of-the-art rate-distortion performance on several commonly used real-world datasets (Figure 1 (e)).

## 2 RELATED WORK

The related work can be divided into two main categories: the compact 3DGS representation and the rate-distortion-optimized 3DGS compression.

**Compact 3DGS Representation.** This category of methods, being a weak form of compression, aims to make more compact the 3DGS representation by pruning, quantizing, or structuring Gaussian primitives. Typical methods that involve pruning include Compact3DGS (Lee et al., 2024), LightGaussian (Fan et al., 2023), EAGLES (Girish et al., 2024), and Trimming the fat (Ali et al., 2024). Compact3DGS features a learnable binary mask and a mask loss to suppress less critical Gaussian primitives during training. In contrast, LightGaussian and EAGLES adopt a post-processing strategy to remove less significant Gaussians based on score-based criteria. In a similar vein, Trimming the fat (Ali et al., 2024) performs pruning progressively. Other methods that involve structuring sparse Gaussian primitives are ScaffoldGS (Lu et al., 2024), OctreeGS (Ren et al., 2024), and F3DGS (Sun et al., 2024). For instance, ScaffoldGS takes an anchor-based approach, where each anchor represents a group of Gaussian primitives whose attributes are represented collectively by a latent feature vector.

**Rate-distortion-optimized 3DGS Compression.** This emerging research area targets the generation and coding of Gaussian primitives in an end-to-end and rate-distortion-optimized manner. Figure 2 presents a taxonomy for methods in this category, including RDO-Gaussian (Wang et al., 2024a), CompGS (Liu et al., 2024), ContextGS (Wang et al., 2024b), and HAC (Chen et al., 2024). Unlike the compact 3DGS representation, these compression techniques involve entropy coding the quantized Gaussian primitives. One central theme is how to predict the probability distributions of the coding features and/or attributes. To this end, some (Chen et al., 2024; Liu et al., 2024; Wang et al., 2024b) borrow the idea of hyperprior from learned image compression to model the distributions of the latent features of Gaussian primitives. One exception is RDO-Gaussian (Wang et al., 2024a), which applies entropy-constrained vector quantization to the Gaussian attributes. As opposed to HAC (Chen et al., 2024) and CompGS (Liu et al., 2024), ContextGS (Wang et al., 2024b) additionally introduces contextual coding in the latent space to leverage the inter correlation between Gaussian primitives. In common, all these schemes consider the hyperprior to be factorial. From Figure 2, our CAT-3DGS represents a novel attempt that makes use of both inter and intra

correlation for coding Gaussian primitives. In terms of the use of inter correlation, it differs from ContextGS (Wang et al., 2024b) and CompGS (Liu et al., 2024) in performing spatial autoregressive coding in the hyperprior domain, which is made possible with our triplane-based hyperprior. More than that, it makes full use of the intra correlation within each individual Gaussian primitive by performing channel-wise autoregressing coding in the latent domain, which is first proposed for 3DGS compression.

## 3 PRELIMINARY

ScaffoldGS (Lu et al., 2024) builds upon 3DGS (Kerbl et al., 2023) and introduces a storage-efficient, anchor-based representation of 3D Gaussian primitives. Instead of directly storing a large number of Gaussian primitives and their attributes, ScaffoldGS introduces the notion of anchor points, with each representing a cluster of Gaussian primitives. The attributes of a Gaussian primitive include its 3D position $\boldsymbol{\mu}^g$, scale $\boldsymbol{s}$, rotation $\boldsymbol{r}$, spherical harmonic coefficients $\boldsymbol{c}$, and opacity $\boldsymbol{\alpha}$. Likewise, each anchor is characterized by its position $\boldsymbol{x}$, latent feature $\boldsymbol{f}$, scaling factor $\boldsymbol{l}$, and $K$ learnable offsets $\{\boldsymbol{O_i}\}_{i=1}^{K}$. The latent feature $\boldsymbol{f}$ encodes the attributes of the Gaussian primitives attached to the same anchor, effectively reducing the data redundancy. The learnable offsets indicate their relative positions with respect to that of the anchor.

With ScaffoldGS, rendering a 2D image involves decoding the view-dependent attributes for all the Gaussians primitives from an anchor representation according to the anchor feature $\boldsymbol{f}$ and camera position $\boldsymbol{x_c}$:

$$\{\boldsymbol{c_i}, \boldsymbol{r_i}, \boldsymbol{s_i}, \boldsymbol{\alpha_i}\}_{i=1}^{K} = F_S(\boldsymbol{f}, \boldsymbol{\sigma_c}, \vec{\boldsymbol{d_c}}), \tag{1}$$

where $\boldsymbol{\sigma_c} = \|(\boldsymbol{x} - \boldsymbol{x_c})\|_2$, $\vec{\boldsymbol{d_c}} = \boldsymbol{x} - \boldsymbol{x_c}/\|\boldsymbol{x} - \boldsymbol{x_c}\|_2$, and $F_S$ is an MLP decoder. The position $\boldsymbol{\mu}_i^g$ of a Gaussian in the cluster is evaluated by adding the anchor position $\boldsymbol{x}$ to the offsets $\boldsymbol{O_i}$, regularized by the scaling factor $\boldsymbol{l}$, as follows:

$$\{\boldsymbol{\mu}_i^g\}_{i=1}^{K} = \boldsymbol{x} + \{\boldsymbol{O_i}\}_{i=1}^{K} \cdot \boldsymbol{l}. \tag{2}$$

Given these parameters, the rendering process proceeds similarly to 3DGS (Kerbl et al., 2023).

## 4 PROPOSED METHOD: CAT-3DGS

Based on ScaffoldGS, this work (termed CAT-3DGS) introduces a content-adaptive triplane approach to coding the anchors' attributes (i.e. the latent feature $\boldsymbol{f} \in \mathbb{R}^{50}$, scaling factor $\boldsymbol{l} \in \mathbb{R}^6$, and offsets $\{\boldsymbol{O_i} \in \mathbb{R}^3\}_{i=1}^{K}$) in an end-to-end, rate-distortion-optimized fashion. Our CAT-3DGS adopts a hyperprior framework to model the probability distributions of the anchors' attributes. Because of the unordered and sparse nature of the anchor points, which collectively form an unorganized point cloud in the 3D space, we project them onto the multi-scale, dense triplanes oriented according to the principal components of the anchor points. As such, our triplane-based hyperprior organizes the projected anchor points in an ordered way on the 2D triplanes. This enables us to use spatial autoregressive models to exploit their *inter correlation* (i.e. spatial correlation) for better entropy coding the hyperprior itself and thus the anchors' attributes. In comparison, the 3D hash-based grid hyperprior (Chen et al., 2024), although compact, is not able to exploit such inherent inter correlation due to the pseudo random mapping between the dense grid points and their hyperprior representations in the hash table. In addition, CAT-3DGS features a channel-wise contextual coding scheme to leverage the *intra correlation* among the components of individual latent features $\boldsymbol{f}$ for their coding. Lastly, we incorporate a view frequency-aware masking mechanism to skip from coding those Gaussian primitives that contribute little to the rendering quality in different views.

### 4.1 SYSTEM OVERVIEW

Figure 3 illustrates our CAT-3DGS framework. The encoding of a 3D scene begins with the generation of the anchor points characterized by their positions $\boldsymbol{x} \in \mathbb{R}^3$ and attributes, including the latent feature $\boldsymbol{f} \in \mathbb{R}^{50}$, offsets $\{\boldsymbol{O_i} \in \mathbb{R}^3\}_{i=1}^{K}$ and scaling $\boldsymbol{l} \in \mathbb{R}^6$. Given the geometry information $\boldsymbol{x}$ of the anchor points, we formulate multi-scale, dense triplanes by conducting a principal component analysis. These triplanes consist of regularly structured grid points, which are quantized and coded by our lightweight spatial autoregressive models (Sec. 4.2). They serve the purpose of

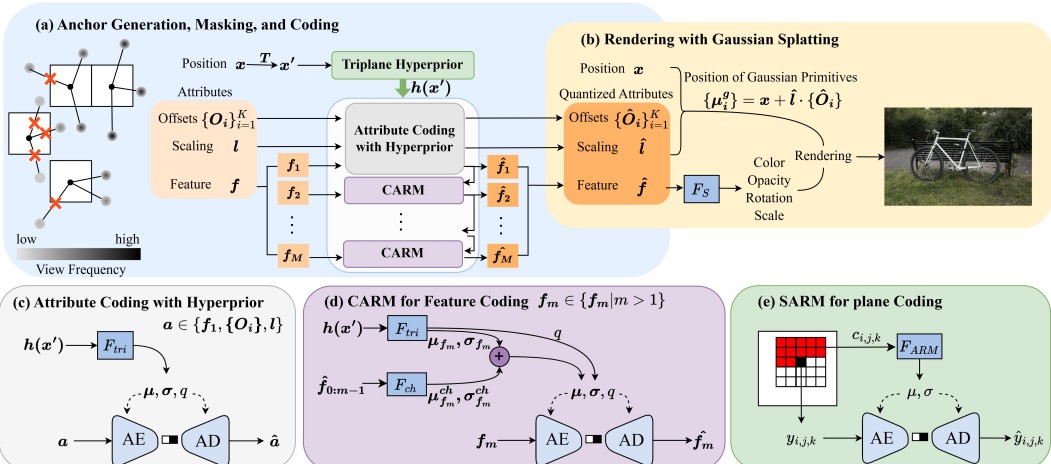

Figure 3: Illustration of our CAT-3DGS framework. CARM: Channel-wise Autoregressive Models. SARM: Spatial Autoregressive Models.

the hyperprior, and are queried and decoded to arrive at the coding distributions of the learned attributes associated with an anchor point according to its position $x$ (Sec. 4.2). Considering that the latent features $f$ collectively constitute a large portion of the compressed bitstream, they each are encoded recursively by channel-wise autoregressive coding (Sec. 4.4). During the learning process, our view frequency-aware masking mechanism is incorporated to mask out the Gaussian primitives which have a minimal impact on the rendering quality (Sec. 4.5). With CAT-3DGS, the information to be compressed in the bitstream include (a) the triplanes, (b) the anchors' attributes and positions, (c) the binary mask, (d) the network weights of the MLP decoder $F_s$, spatial autoregressive model $F_{ARM}$, hyperprior decoder $F_{tri}$, and channel-wise autoregressive model $F_{ch}$. The anchors' positions $x$ and the network weights are signaled in 16-bit and 32-bit floating-point formats, respectively. The binary mask is entropy encoded.

The rendering of a 2D image proceeds in much the same way as Scaffold-GS (Lu et al., 2024). Specifically, we use an MLP decoder $F_S$ to decode the coded latent feature $\hat{f}$ of an anchor point to obtain the attributes (i.e. color, opacity, rotation, scale) of all the Gaussian primitives belonging to the same anchor. Accordingly, the coded offsets $\{\hat{O_i}\}_{i=1}^K$ and scaling $\hat{l}$ are combined with the anchor's position $x$ to reconstruct their positions. The same decoding process is repeated for the remaining anchor points needed to render the 2D image of a specific viewpoint.

## 4.2 TRIPLANE-BASED HYPERPRIOR

Our triplane-based hyperprior aims to learn the prior distributions on the attributes (the latent features $f$, offsets $\{O_i\}_{i=1}^K$ and scaling $l$) of the anchor points. Conceptually, a triplane is composed of three 2D planes, denoted as $\mathcal{P}_c, c \in \{xy, yz, zx\}$, of the same 2D spatial resolution and channel dimension (See Figure 1 (b)). The notion of triplanes originates from decomposing a dense, 3D grid, which is costly to represent, into three 2D planes, which are storage friendly yet with more restricted expressiveness. Each grid point in these 2D planes is a learnable parameter. Our CAT-3DGS learns multiple triplanes of various resolutions to capture both coarse and fine detail. We thus augment $\mathcal{P}_c$ with an upsampling scale $r$ as $\mathcal{P}_{r,c} \in \mathbb{R}^{ch \times rB \times rB}$, where $ch$ denotes the number of channels, $r$ are integers denoting the upsampling scales and $B$ denotes the spatial resolution of the triplane at the lowest scale (i.e. $r = 1$).

To retrieve the hyperprior $h(x)$ for an anchor point $x$ in the 3D space, we project $x$ onto each 2D plane $\mathcal{P}_{r,c}$, with the projected 2D coordinates given by $\pi_{r,c}(x)$. When $\pi_{r,c}(x)$ is fractional, we interpolate between the nearest integer grid points with an interpolation kernel $\psi$ to get the triplane feature. In symbols, we have $\psi(\mathcal{P}_{r,c}, \pi_{r,c}(x))$. The same process is repeated for every combination of permissible $r$ and $c$, with the resulting triplane features concatenated to formulate the hyperprior

$h(\boldsymbol{x})$:

$$h(\boldsymbol{x}) = \bigcup_r \bigcup_{c \in \{xy, yz, zx\}} \psi(\boldsymbol{\mathcal{P}}_{r,c}, \pi_{r,c}(\boldsymbol{x})). \tag{3}$$

In doing so, we notice that $\boldsymbol{x}$ may potentially be unbounded. However, the spatial resolutions of the triplanes must be bounded, because these triplanes need to be signaled in the bitstream. With their finite spatial resolutions, a contraction function (Barron et al., 2022b) is applied in order to fit the potentially unbounded $\boldsymbol{x}$ to our bounded triplanes:

$$\text{contract}(\boldsymbol{x}) = \begin{cases} \boldsymbol{x} & \text{if } \|\boldsymbol{x}\| \le 1 \\ (2 - \frac{1}{\|\boldsymbol{x}\|})(\frac{\boldsymbol{x}}{\|\boldsymbol{x}\|}) & \text{if } \|\boldsymbol{x}\| > 1. \end{cases} \tag{4}$$

Careful examination of Eq. (4) reveals that $\boldsymbol{x}$ needs to be normalized in order to minimize the impact of the non-linear scaling applied to $\boldsymbol{x}$ with $\|\boldsymbol{x}\| > 1$ while maximizing the usage of the grid points in each triplane to represent those $\boldsymbol{x}$ with $\|\boldsymbol{x}\| \le 1$. To this end, we conduct a principal component analysis (PCA) with respect to the positions $\boldsymbol{x}$ of the anchor points, performing a linear transformation $T(\boldsymbol{x})$ of $\boldsymbol{x}$ before it is contracted with Eq. (4). More specifically, $T(\boldsymbol{x})$ is given by

$$\boldsymbol{x}' = T(\boldsymbol{x}) = \text{contract}(\frac{\boldsymbol{R}^{\boldsymbol{x}}(\boldsymbol{x} - \boldsymbol{\mu}^{\boldsymbol{x}})}{\boldsymbol{\sigma}^{\boldsymbol{x}}}), \tag{5}$$

where $\boldsymbol{\mu}^{\boldsymbol{x}} \in \mathbb{R}^3$ is the mean vector, $\boldsymbol{\sigma}^{\boldsymbol{x}} \in \mathbb{R}^3$ are the variances along the three principal axes, and $\boldsymbol{R}^{\boldsymbol{x}} \in \mathbb{R}^{3 \times 3}$ is the PCA rotation matrix. As illustrated in Figure 1 (d), this transformation centers the point cloud of the anchor points, allowing most of the central anchor points to be linearly scaled. We finally substitute $T(\boldsymbol{x})$ into Eq. (3) to evaluate the hyperprior $h(\boldsymbol{x}')$.

To entropy encode (or decode) the quantized attributes $\hat{\boldsymbol{a}} \in \{\hat{\boldsymbol{f}}_1, \{\hat{\boldsymbol{O}}_i\}, \hat{\boldsymbol{l}}\})$ of an anchor, an MLP $F_{tri}$ is used to decode $h(\boldsymbol{x}')$ to predict their means, variances, and quantization step size. That is, $(\boldsymbol{\mu}, \boldsymbol{\sigma}, q) = F_{tri}(h(\boldsymbol{x}'))$. Notably, each of these attributes is assumed to follow a Gaussian distribution, with their coding probabilities given by

$$p(\hat{\boldsymbol{a}} | h(\boldsymbol{x}')) = \int_{\hat{\boldsymbol{a}} - \frac{q}{2}}^{\hat{\boldsymbol{a}} + \frac{q}{2}} \mathcal{N}(\boldsymbol{\mu}, \boldsymbol{\sigma}) \, d\boldsymbol{a}. \tag{6}$$

The acute reader may have observed that only part of the latent feature $\hat{\boldsymbol{f}}$ is involved in Eq. (6). The coding of the remaining part (i.e. $\hat{\boldsymbol{f}}_2, \hat{\boldsymbol{f}}_3, ...$) will be elaborated in Sec. 4.4.

### 4.3 SPATIAL AUTOREGRESSIVE MODELS (SARM) FOR TRIPLANE CODING

The triplane-based hyperprior must be encoded into the bitstream. Observing that the grid points in each 2D plane capture to a large extent the *inter correlation* (i.e. spatial correlation) between the anchor points in the 3D space, we introduce spatial autoregressive models for triplane coding. Currently, a dedicated autoregressive model $F_{ARM}$ is learned and shared for 2D planes $\boldsymbol{\mathcal{P}}_{r,c}$ of the same orientation $c \in \{xy, yz, zx\}$ without regard to its upsampling scale $r$. Thus, a total of three $F_{ARM}$, one for each orientation, are learned. Moreover, the 2D plane $\boldsymbol{\mathcal{P}}_{r,c}$ has $ch$ channels. These channels are encoded (and decoded) independently of each other with the same $F_{ARM}$ to strike a balance between complexity and coding efficiency. In fact, all the channels from these 2D planes $\boldsymbol{\mathcal{P}}_{r,c}$ can be encoded (and decoded) in parallel.

To entropy encode (and decode) a grid point $y_{i,j,k}$ in the channel $k$ of the 2D plane $\boldsymbol{\mathcal{P}}_{r,c}$, $F_{ARM}$ formulates the context $c_{i,j,k}$ by referring to the previously decoded grid points of the same channel in the neighborhood specified by $c_{i,j,k} = [\hat{y}_{i-2:i-1, j-2:j+2, k}; \hat{y}_{i,j-2:j-1,k}]$ (Figure 3 (e)). It outputs the parameters $(\mu_{i,j,k}, \sigma_{i,j,k}) = F_{ARM}(c_{i,j,k})$ of a Laplace distribution that models the distribution of $y_{i,j,k}$. The coding probability of the quantized grid point $\hat{y}_{i,j,k}$ is then given by

$$p(\hat{y}_{i,j,k} | c_{i,j,k}) = \int_{\hat{y}_{i,j,k} - \frac{Q}{2}}^{\hat{y}_{i,j,k} + \frac{Q}{2}} Laplace(\mu_{i,j,k}, \sigma_{i,j,k}) \, dy_{i,j,k}, \tag{7}$$

where $Q = 1/16$ is the quantization step size. This small quantization step is chosen to make the training more stable when the training process transitions from the non-quantization-aware training to the quantization-aware training.

### 4.4 Channel-wise Autoregressive Models (CARM) for Feature Coding

The coding of the latent features $\boldsymbol{f}$ deserves additional effort as they normally represent a considerable portion of the compressed bitstream. For efficient feature coding, we leverage the *intra correlation* among the components of a feature $\boldsymbol{f}$. We divide every individual feature $\boldsymbol{f}$ into $M$ slices along the channel dimension, followed by introducing a channel-wise autoregressive models for coding these slices. The coding of slice $\boldsymbol{f_m}$ with $m > 1$ is able to benefit from referring to the previous coded slices, i.e. $\{\boldsymbol{f_i} | i < m\}$, and the hyperprior. In symbols, we have the coding probability of the quantized slice $\hat{\boldsymbol{f}}_m$ as

$$p(\hat{\boldsymbol{f}}_m | h(\boldsymbol{x'}), \hat{\boldsymbol{f}}_{0:m-1}) = \int_{\hat{\boldsymbol{f}}_m - \frac{q}{2}}^{\hat{\boldsymbol{f}}_m + \frac{q}{2}} N(\boldsymbol{\mu} + \boldsymbol{\mu_{ch}}, \boldsymbol{\sigma} + \boldsymbol{\sigma_{ch}}) d\boldsymbol{f}, \tag{8}$$

where $(\boldsymbol{\mu}, \boldsymbol{\sigma}, q) = F_{tri}(h(\boldsymbol{x'}))$ and $(\boldsymbol{\mu_{ch}}, \boldsymbol{\sigma_{ch}}) = F_{ch}(\hat{\boldsymbol{f}}_{0:m-1})$. In our design, the slices are unevenly partitioned, with further details provided in Sec. 5.3.

### 4.5 View Frequency-Aware Masking

Our view frequency-aware masking is designed to distinguish between Gaussian primitives in terms of their potential contribution to the rendering quality. As observed in Compact-3DGS (Lee et al., 2024) and HAC (Chen et al., 2024), using a learnable binary mask $M_{n,k}$ to mask out the $k$-th Gaussian primitive of the $n$-th anchor can be effective in reducing the number of Gaussian primitives to be signaled, thereby saving the storage space and transmission bandwidth. This masking mechanism is usually implemented as $M_{n,k} = \mathbb{1}(sigmoid(m_{n,k}) > \epsilon)$, where $m_{n,k}$ is a learnable parameter and $\epsilon$ is a global hyperparameter shared across every Gaussian primitive to determine its existence. Although effective, this blind approach may risk removing some critical Gaussian primitives. A question that arises naturally is whether we could prioritize Gaussian primitives in the masking process according to their potential contribution to the rendering quality. We observe that during training, some Gaussian primitives are more frequently used in rendering the training views. As such, we attach to each Gaussian primitive a weight $p_{n,k}$ that reflects its relative frequency of being used in rendering these training views. We adopt $p_{n,k}$ in our masking function:

$$M_{n,k} = \mathbb{1}(sigmoid(m_{n,k}) \cdot p_{n,k} > \epsilon). \tag{9}$$

With the same $\epsilon$ applied to every Gaussian primitive, a higher $p_{n,k}$ requires $sigmoid(m_{n,k})$ to approach zero more closely in order to skip the corresponding Gaussian primitive, making the task more difficult. As a result, more Gaussian primitives that are critical to rendering the training views are retained. For this scheme to work well, the basic premise is that the distribution of training views should be similar to that of test views. We argue that this is true to some extent because when their distributions differ significantly, there is little guarantee of the rendering quality in those test views.

### 4.6 Training Objectives

The training of CAT-3DGS involves minimizing the rate-distortion cost $L_{\text{Scaffold}} + \lambda_r L_{\text{rate}}$ together with a masking loss $\lambda_m L_m$:

$$L = L_{\text{Scaffold}} + \lambda_r L_{\text{rate}} + \lambda_m L_m, \tag{10}$$

where we follow Scaffold-GS Lu et al. (2024) to evaluate $L_{\text{Scaffold}}$, which includes the distortion between the original and rendered images as well as a regularization term imposed on the scales $s$ of Gaussian primitives. $L_{\text{rate}}$ indicates the number of bits needed to signal the hyperprior and the anchors' attributes:

$$L_{\text{rate}} = \frac{1}{N(50 + 6 + 3K)}(L_{\text{rate}}^{\mathcal{A}} + \lambda_{\text{tri}} L_{\text{rate}}^{\mathcal{P}}), \tag{11}$$

where $L_{\text{rate}}^{\mathcal{A}} = -\sum_{\hat{a}} log_2 \, p(\hat{a})$ is the estimated bit rate of the anchors' attributes, $L_{\text{rate}}^{\mathcal{P}} = -\sum_{\hat{y}} log_2 \, p(\hat{y})$ is the triplanes' bit rate, and $N(50 + 6 + 3K)$ is the total number of parameters of anchors' attributes. In Eq. (10), $L_m$ is the mask loss adopted from Compact3DGS (Lee et al., 2024) to regularize the view frequency-aware masking:

$$L_m = \sum_{n=1}^{N} \sum_{k=1}^{K} sigmoid(m_{n,k}). \tag{12}$$

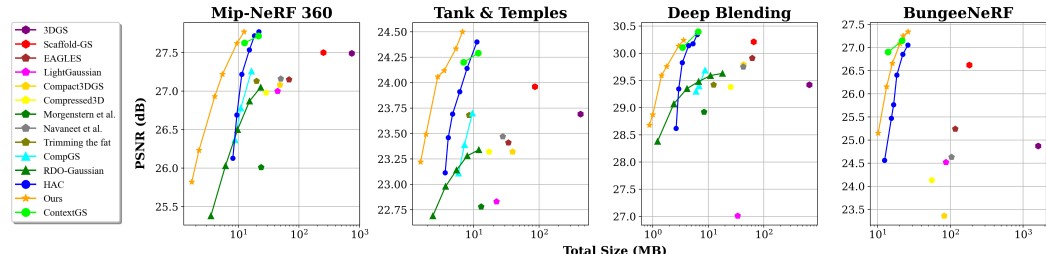

Figure 4: Rate-distortion comparison of our CAT-3DGS, HAC, ContextGS, RDO-Gaussian, CompGS, and several other compact 3DGS representations (normally without entropy coding and visualized as rate-distortion points).

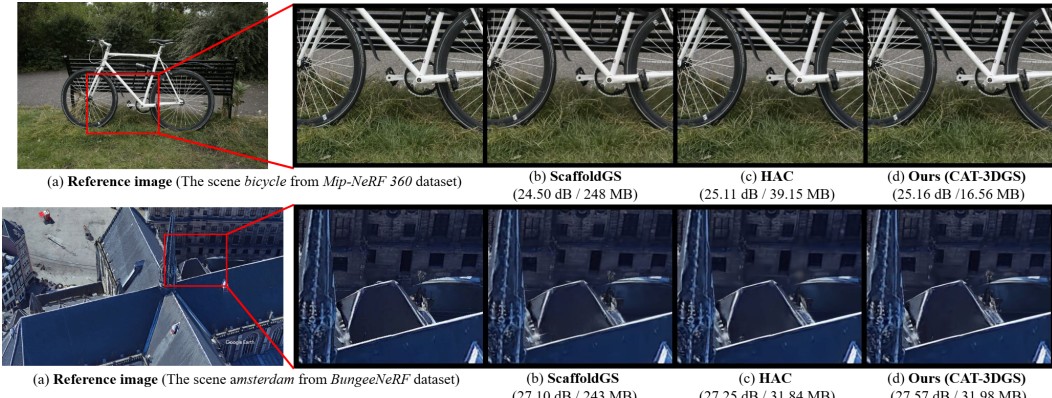

(a) **Reference image** (The scene *bicycle* from *Mip-NeRF 360* dataset)   (b) **ScaffoldGS** (24.50 dB / 248 MB)   (c) **HAC** (25.11 dB / 39.15 MB)   (d) **Ours (CAT-3DGS)** (25.16 dB /16.56 MB)

(a) **Reference image** (The scene a*msterdam* from *BungeeNeRF* dataset)   (b) **ScaffoldGS** (27.10 dB / 243 MB)   (c) **HAC** (27.25 dB / 31.84 MB)   (d) **Ours (CAT-3DGS)** (27.57 dB / 31.98 MB)

Figure 5: Qualitative results of our CAT-3DGS, HAC and ScaffoldGS.

In particular, $\lambda_m = \max(10^{-3}, 0.3 \cdot \lambda_r)$ changes with the rate parameter $\lambda_r$. Further details about this design aspect are provided in Appendix B.

## 5 EXPERIMENTAL RESULTS

### 5.1 IMPLEMENTATION DETAILS

This part summarizes some crucial implementation details for reproduciability. First, the spatial resolution $B$ of the triplane at the lowest scale ($r = 1$) is determined in proportional to the number of anchor points obtained after 10k training iterations. The choices of the other hyperparameters include: the channel number $ch = 72$, $\epsilon = 0.01$ (0.0004 for BungeeNeRF) for the view frequency-aware masking, $M = 4$ with uneven slices $(5, 10, 15, 25)$ for the channel-wise autoregressive coding. The rate parameter $\lambda_r$ ranges from 0.002 to 0.04, and from 0.001 to 0.02 for BungeeNeRF. Lastly, our triplanes have only two scales; that is, $r = 1, 2$.

### 5.2 RATE-DISTORTION COMPARISON

**Baselines.** For comparison, our baseline methods include (1) the vanilla 3DGS, (2) ScaffoldGS (our base model), and (3) four rate-distortion-optimized approaches–namely, HAC (Chen et al., 2024), RDO-Gaussian (Wang et al., 2024a), CompGS (Liu et al., 2024), and ContextGS (Wang et al., 2024b). Notably, ContextGS is a concurrent work of our CAT-3DGS. Due to the emerging nature of the rate-distortion-optimized 3DGS compression, there are only few early attempts. We thus also include for comparison several compact 3DGS techniques without joint rate-distortion-optimized training (Lee et al., 2024; Fan et al., 2023; Niedermayr et al., 2024; Navaneet et al., 2023; Morgenstern et al., 2023; Girish et al., 2024; Ali et al., 2024). Generally, these techniques do not consider entropy coding. They are visualized as individual rate-distortion points in our rate-distortion plots.

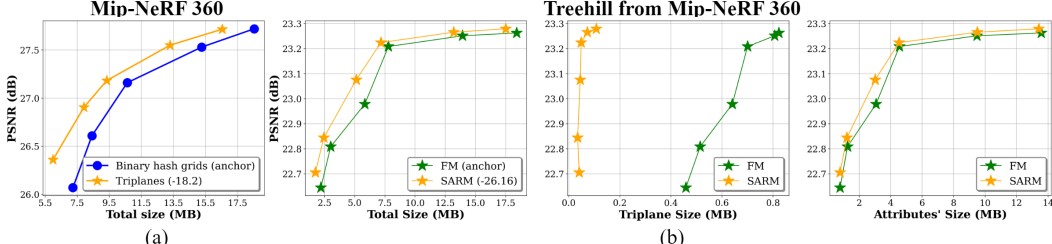

Figure 6: (a) Rate-distortion curves comparing our triplanes and the binary hash grids in HAC. (b) Rate-distortion curves comparing our SARM with the factorized model (FM).

**Datasets.** We follow the common test protocol to test our CAT-3DGS on real-world scenes, including Mip-NeRF 360 (Barron et al., 2022a), Tanks & Temples (Knapitsch et al., 2017), Deep Blending (Hedman et al., 2018) and BungeeNeRF (Xiangli et al., 2022). For comparison, we choose the same scenes from each dataset as those used in the prior works (Lu et al., 2024; Chen et al., 2024).

**Metrics.** We compare the rate-distortion performance of the competing methods by visualizing their rate-distortion plots. The quality metric is PSNR measured in the RGB domain. The bit rate is the file size of the compressed bitstream obtained by performing entropy encoding. When reporting the rate-distortion results for a dataset, we take the average of the per-sequence PSNRs and file sizes. In our ablation study, we additionally report the BD-rate saving (Bjontegaard, 2001) to single out the contribution of individual components. In particular, the BD-rate saving is evaluated for each test scene and averaged across all the scenes in the dataset. Negative values suggest rate saving at the same quality level as compared to the anchor (a chosen baseline method) and vice versa. Note that evaluating the BD-rate requires at least 4 rate-distortion points and largely overlapping distortion intervals. We thus use it only in our ablation study.

**Compression Results.** In Figure 4, our CAT-3DGS outperforms the competing methods (particularly those rate-distortion-optimized ones) across all the datasets, achieving the state-of-the-art rate-distortion performance. On the Mip-NeRF 360 dataset, our CAT-3DGS achieves (at its second highest rate point) 78× and 26x rate reductions than 3DGS and ScaffoldGS, respectively, while achieving slightly higher PSNR by 0.16 dB. Figure 5 offers the subjective quality comparison among ScaffoldGS, HAC, and our CAT-3DGS. On the *bicycle* scene, our CAT-3DGS achieve a 57% size reduction while showing similar visual quality to HAC. Likewise, on the *amsterdam* scene, it achieves 0.32dB higher PSNR and better subjective quality than HAC, but with a similar file size.

## 5.3 ABLATION EXPERIMENTS

We conduct ablation experiments on the Mip-NeRF360 dataset for its diverse scenes.

**Triplanes versus Binary Hash Grids.** This study investigates the benefits of our triplane-based hyperprior. Based on the HAC framework, we change its hyperprior from the binary hash grids to our multi-scale triplanes with spatial autoregressive coding. The remaining components and training procedure are the same as HAC. From Figure 6 (a), our triplane-based hyperprior achieves an 18% BD-rate saving. It highlights the advantage of the triplane representations, which are able to capture the spatial correlation of the anchor points and enable more efficient entropy coding with spatial autoregressive models.

**Spatial Autoregressive Models (SARM) versus Factorized Models (FM).** Based on CAT-3DGS, this ablation study replaces our SARM with FM. The latter assumes that the triplane-based hyperprior has independent and identically distributed components. For fair comparison, 3 FMs (one for each plane orientation) are trained and used for coding the triplanes. The result on Mip-NeRF360 indicates that our SARM achieves 19% BD-rate saving as compared to FM. The result suggests the strong spatial correlation in the triplane-based hyperprior. Figure 6 (b) offers a breakdown analysis, showing that SARM benefits not only the coding of the triplane-based hyperprior, but more importantly that of the anchors' attributes, which constitute the major portion of the compressed bitstream.

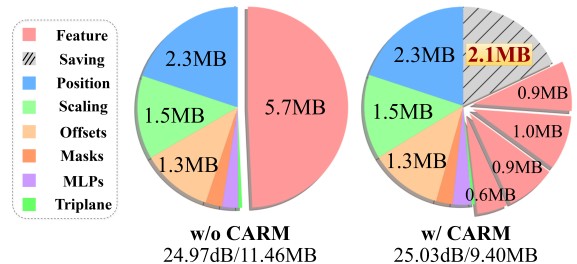

Figure 7: Breakdown analysis of different coding parts w/ and w/o our CARM for the *bicycle* scene.

Table 1: The impact of the slice number and partition in our CARM on compression performance The results are obtained with Mip-NeRF 360.

| $M$ Slices | Channels per Slice | BD-rate |
|---|---|---|
| 1 | 50 | 0 |
| 2 | 25, 25 | -6.3 |
| 2 | 15, 35 | -8.9 |
| 4 | 12, 12, 13, 13 | -8.9 |
| 4 | 5, 10, 15, 20 | -11.9 |

Table 2: Comparison of decoding time and rendering throughput: Ours (CAT-3DGS) vs. HAC.

| | Scene | Anchor Count (K) | Base Resolution $B$ | Decoding Time (s) ↓ | | | Rendering Speed (FPS) ↑ |
|---|---|---|---|---|---|---|---|
| | | | | Triplane | Anchor Attributes | Total | |
| **Ours** | room | 36.5 | 64 | 11.4 | 2.2 | 13.6 | 127.0 |
| | amsterdam | 533.0 | 128 | 47.4 | 17.0 | 64.4 | 83.4 |
| **HAC** | room | 228.8 | N/A | N/A | - | 6.6 | 103.1 |
| | amsterdam | 483.5 | N/A | N/A | - | 11.3 | 77.9 |

**Channel-wise Autoregressive Models (CARM).** Figure 7 presents two pie charts to single out the contribution of our CARM on the *bicycle* scene. As shown, CARM reduces the compressed size of the latent features from 5.7 MB to 3.4 MB, amounting to a 40% rate reduction in this part of the bitstream. We also observe a similar trend in the other scenes. More results are provided in Appendix F. From Table 1, more slices lead to improved coding performance. Unevenly-partitioned slices perform slightly better than evenly-partitioned slices. With uneven slices, we found that more essential information is packed in the first or two slices, an effect that is similar to energy compaction and is much desirable for coding purposes. More discussions are presented in Appendix C.

**View Frequency-Aware Masking.** We conduct another ablation study that disables the view frequency-aware masking in our CAT-3DGS. In other words, we remove the weight $p_{n,k}$ in the masking function. Doing so results in a 16% BD-rate drop on Mip-NeRF360. The removal of Gaussian primitives less critical to the rendering quality helps reduce the bit rate. More results are provided in Appendix D.

**Decoding Time and Rendering Throughput.** Table 2 compares the decoding time (seconds) and rendering throughput (frames per second) of our CAT-3DGS and HAC for two scenes, *amsterdam* in BungeeNeRF (Xiangli et al., 2022) and *room* in Mip-NeRF360 (Barron et al., 2022b). This information is collected on one NVIDIA V100. CAT-3DGS has much higher decoding time than HAC due to the use of autoregressive models. However, in terms of rendering throughput, our CAT-3DGS is faster than HAC. This is because our frequency-aware masking effectively reduces the number of anchors. The decoding time of CAT-3DGS can be further improved by making full use of the parallelism in decoding triplanes. Recall that all the 2D planes in our triplane-based hyerprior are independently decodable and so are the channels in each 2D plane (Sec. 4.3). Currently, only the channel parallelism is used in our implementation. The decoding of different anchor attributes is parallelizable to some extent. The coding dependency is in the channel dimension and not between the anchors' attributes. Last but not least, the 3DGS system normally has decoding and rendering as two decoupled processes. The Gaussian primitives are decoded first, followed by rendering images in different views. The decoding is generally less time sensitive and has little impact on the rendering process, which is the same as ScaffoldGS (Lu et al., 2024) with our CAT-3DGS.

## 6 CONCLUSIONS

This work presents a novel rate-distortion-optimized 3DGS compression framework. In an effort to leverage the inter correlation between Gaussian primitives for coding, it features PCA-guided triplanes as the hyperprior and incorporates a spatial autoregressive model for their coding. Furthermore, a channel-wise autoregressive model is introduced for the first time to explore the intra correlation within each individual Gaussian primitive for coding. When combined with a view frequency-aware masking mechanism, these features lead to the state-of-the-art coding performance.

ACKNOWLEDGMENTS

This work is supported by National Science and Technology Council, Taiwan under the Grant NSTC 113-2634-F-A49-007-, MediaTek, and National Center for High-performance Computing, Taiwan.

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

# A  TRAINING PROCESS

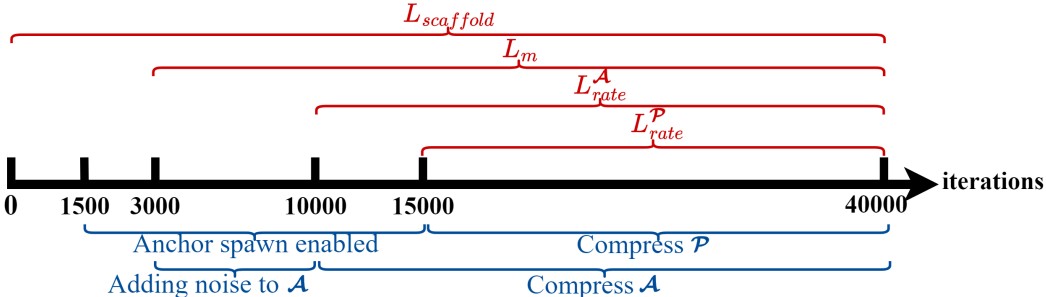

Figure 8: Detailed Training Process of our CAT-3DGS.

Figure 8 depicts our detailed training procedure. It includes the following training stages.

**Anchor Spawning**  In this stage, we adopt ScaffoldGS to ensure a stable start of both anchor attribute training and anchor spawning. However, to simulate the quantization effect, we introduce noise to the attributes of the anchor points starting at iteration 3000. To prevent the generation of an excessive number of anchors, we disable anchor growing during iterations 3000 to 4000.

**Triplane-based Hyperprior**  After adding noise to the attributes of the anchor points during iterations 3000 to 10,000, we begin using the triplane as a hyperprior to learn the distribution of anchors' attributes starting from the 10,000th iteration.

**Spatial Autoregressive Models for Triplane Coding**  We start triplane coding at iteration 15,000. Specifically, we warm up the spatial autoregressive model while freezing the other learnable parameters between iterations 15,000 and 16,000.

# B  RATE-AWARE MASK TRADE-OFF (RMT)

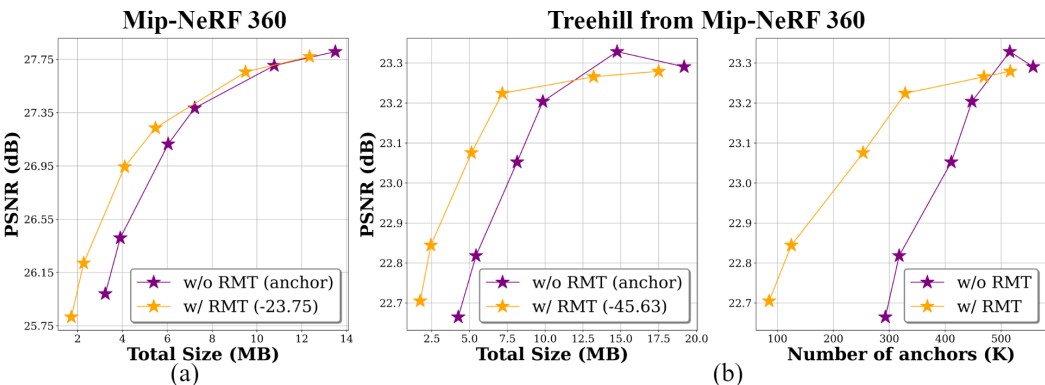

Figure 9: Rate-distortion comparison w/ and w/o our rate-aware mask trade-off. RMT: Rate-aware Mask Trade-off.

Based on the observation that the number of anchors can be further reduced while maintaining similar PSNR, we accordingly relate the mask hyperparameter $\lambda_m$ to the rate hyperparameter $\lambda_r$ using the relationship $\lambda_m = \max(10^{-3}, 0.3 \cdot \lambda_r)$. This implies that, at higher bit rates, $\lambda_r$ decreases and more offsets (i.e. Gaussian primitives) and anchors are kept. Conversely, at lower bit rates, $\lambda_r$ increases and more offsets and anchors are removed.

In Figure 9 (a), we evaluate the performance with a fixed mask trade-off set to 0.0005 (labeled "w/o RMT") and compare it with the rate-aware mask trade-off approach (labeled "w/ RMT") on Mip-

NeRF 360. The rate-aware mask trade-off has greater coding performance gain, particularly at lower bit rates. For further details regarding this gain, we select the *treehill* scene from Mip-NeRF 360, as shown in Figure 9 (b). Our method with RMT demonstrates that, at the lowest rate point, the number of anchor points is only one-third of that "w/o RMT", and the total size achieves a 60% reduction, while maintaining similar PSNR.

## C THE IMPACT OF SLICE PARTITIONING ON OUR CARM

Table 3: Analysis the number of bits per channel within a slice in different cases.

| Method | slices | Channels per slice | Size per channel within one slice (kB) | Size for $f$ (MB) | Total size (MB) | PSNR |
|---|---|---|---|---|---|---|
| w/o CARM | 1 | (50) | (8.91) | 0.44 | 1.69 | 30.78 |
| w/ CARM (even) | 4 | (12, 12, 13, 13) | (22.2, 6.38, 4.65, 3.98) | 0.45 | 1.74 | 30.85 |
| w/ CARM (uneven) | 4 | (5, 10, 15, 20) | (17.22, 9.61, 5.35, 3.64) | 0.33 | 1.59 | 30.89 |

In this section, we further analyze the number of bits per channel within a slice in the cases of w/o CARM, w/ CARM (even), and w/ CARM (uneven), as shown in Table 3. We observe that when only one slice is used, indicating CARM is disabled, the kilo bytes per channel is 8.91. However, when CARM is enabled, the first slice uses 22.2 kilo bytes per channel for the even partition case and 17.22 kilo bytes per channel for the uneven case. The data suggest that the earlier slices contain more information, resulting in larger sizes. This also indicates that the richer information in the earlier slices can help the coding of the subsequent slices, allowing them to use fewer bits. As for the even and uneven cases, we observe that the uneven partition yields a better result, reducing the size of $f$ by 0.12MB compared to the even case.

## D VIEW FREQUENCY-AWARE MASKING

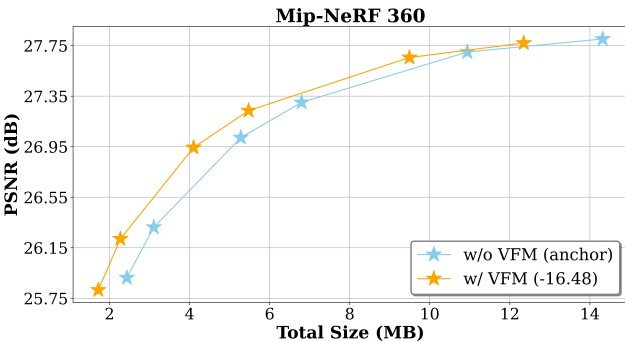

Figure 10: Rate-distortion comparison w/ and w/o our view frequency-aware masking. VFM: View Frequency-aware Masking.

In Figure 10, we compare CAT-3DGS (denoted by "w/ VFM") with a variant (denoted by "w/o VFM") that removes the weight $p_{n,k}$ in the masking function. The view frequency-aware masking has a significant effect at low bit rates, reducing the total size by 29% at the lowest rate point while PSNR drops by only 0.1dB. At the highest rate point, it reduces the total size by 14% while still maintaining similar PSNR. This indicates that the view frequency-aware masking can retain more important points while pruning a larger number of relatively less important ones, even when most of the anchors are masked out.

## E BITSTREAM OF EACH COMPONENT

Our bit stream consists of seven components: the anchor positions $a$, three anchor attributes (features $f$, offsets $\{O_i\}$, and scaling factors $l$), a set of triplanes $\mathcal{P}$, binary masks $M$, and MLPs $F_S$, $F_{tri}$, $F_{ch}$, $F_{ARM}$. After applying our triplane hyperprior and channel-wise autoregressive model, the size of the attributes has been significantly reduced. Additionally, due to the spatial autoregressive

model, the triplane size occupies only a small portion. The rate-aware mask tradeoff further allows us to reduce the number of anchors that need to be compressed at low bit rates, thereby reducing the total size.

Table 4: Bitstream of each component. The result is for the scene *treehill* on Mip-NeRF 360 dataset.

| | Number of Anchors (K) | Bitstream of Each Component (MB) | | | | | | | Total size (MB) | Fidelity | |
|---|---|---|---|---|---|---|---|---|---|---|---|
| | | Position | Feature | Scaling | Offsets | Masks | MLPs | Triplane | | PSNR | SSIM |
| treehill (high-rate) | 516.3 | 3.10 | 7.75 | 2.47 | 3.19 | 0.53 | 0.35 | 0.11 | 17.5 | 23.28 | 0.646 |
| treehill (low-rate) | 85.1 | 0.51 | 0.33 | 0.28 | 0.18 | 0.06 | 0.35 | 0.04 | 1.74 | 22.71 | 0.558 |

## F  ADDITIONAL BREAKDOWN RESULTS FOR CARM

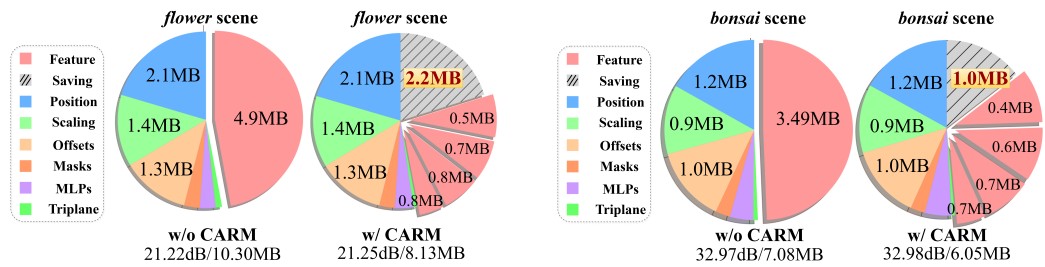

Figure 11: The breakdown analyses of different coding parts w/ and w/o our CARM for the *flower* and *bonsai* scenes, respectively.

Figure 11 provides additional breakdown results for *flower* and *bonsai* scenes. The results confirm again that our CARM effectively reduces the compressed size of the latent features associated with anchors' attributes.

## G  THE IMPACT OF CARM ON DECODING TIME

Table 5: Comparison of the decoding time w/ and w/o our CARM.

| Scene | Method | Decoding Time (s) | | |
|---|---|---|---|---|
| | | Triplane | Anchor Attributes | Total |
| room | CAT-3DGS (w/ CARM) | 11.4 | 2.2 | 13.6 |
| | CAT-3DGS w/o CARM | 11.3 | 2.1 | 13.4 |
| amsterdam | CAT-3DGS (w/ CARM) | 47.4 | 17.0 | 64.4 |
| | CAT-3DGS w/o CARM | 47.2 | 14.9 | 62.1 |

We compare the decoding time of our schemes w/ and w/o CARM in Table 5. The results indicate that CARM has a negligible impact on the decoding time.

## H  COMPLETE BREAKDOWN OF QUANTITATIVE RESULTS

**Quantitative results**  For a more comprehensive data presentation, we present detailed information on the rate-distortion curves, as shown in Figure 4, and the quantitative results are shown in Table 6.

**Per-scene Results of Our CAT-3DGS Framework**  The detailed results of our approach for Mip-NeRF 360 dataset (Barron et al., 2022a) are presented in Table 7.

The detailed results of our approach for BungeeNeRF dataset (Xiangli et al., 2022) are presented in Table 8.

The detailed results of our approach for Tank&Temples dataset (Knapitsch et al., 2017) are presented in Table 9.

The detailed results of our approach for Deep Blending dataset (Hedman et al., 2018) are presented in Table 10.

**Per-scene Results of the Baseline Models**    Per-scene results for all datasets from our two baseline models, ScaffoldGS (Lu et al., 2024) and HAC (Chen et al., 2024) are also provided in Table 15 and Table 11 12 13 14, respectively.

Table 6: The Quantitative results of our CAT-3DGS and other approaches. 3DGS (Kerbl et al., 2023) and ScaffoldGS (Lu et al., 2024) are baseline methods, which are presented in the first section. Approaches in the second section are compact representation, while the third section covers rate-distortion-optimized compression approaches. For comparison, we also provide two results of different size and fidelity tradeoffs by adjusting $\lambda_r$. The size is measured in megabytes (MB).

| Datasets Methods | Mip-NeRF360 | | | | Tank&Temples | | | | DeepBlending | | | | BungeeNeRF | | | |
|---|---|---|---|---|---|---|---|---|---|---|---|---|---|---|---|---|
| | psnr ↑ | ssim ↑ | lpips ↓ | size ↓ | psnr ↑ | ssim ↑ | lpips ↓ | size ↓ | psnr ↑ | ssim ↑ | lpips ↓ | size ↓ | psnr ↑ | ssim ↑ | lpips ↓ | size ↓ |
| 3DGS(SIGGRAPH'23) | 27.49 | 0.813 | 0.222 | 744.7 | 23.69 | 0.844 | 0.178 | 431.0 | 29.42 | 0.899 | 0.247 | 663.9 | 24.87 | 0.841 | 0.205 | 1616 |
| ScaffoldGS (CVPR'24) | 27.50 | 0.806 | 0.252 | 253.9 | 23.96 | 0.853 | 0.177 | 86.50 | 30.21 | 0.906 | 0.254 | 66.00 | 26.62 | 0.865 | 0.241 | 183.0 |
| EAGLES(ECCV'24) | 27.15 | 0.808 | 0.238 | 68.89 | 23.41 | 0.840 | 0.200 | 34.00 | 29.91 | 0.910 | 0.250 | 62.00 | 25.24 | 0.843 | 0.221 | 117.1 |
| LightGaussian | 27.00 | 0.799 | 0.249 | 44.54 | 22.83 | 0.822 | 0.242 | 22.43 | 27.01 | 0.872 | 0.308 | 33.94 | 24.52 | 0.825 | 0.255 | 87.28 |
| Compact3DGS (CVPR'24) | 27.08 | 0.798 | 0.247 | 48.80 | 23.32 | 0.831 | 0.201 | 39.43 | 29.79 | 0.901 | 0.258 | 43.21 | 23.36 | 0.788 | 0.251 | 82.60 |
| Compressed3D (CVPR'24) | 26.98 | 0.801 | 0.238 | 28.80 | 23.32 | 0.832 | 0.194 | 17.28 | 29.38 | 0.898 | 0.253 | 25.30 | 24.13 | 0.802 | 0.245 | 55.79 |
| Morgenstern et al. | 26.01 | 0.772 | 0.259 | 23.90 | 22.78 | 0.817 | 0.211 | 13.05 | 28.92 | 0.891 | 0.276 | 8.40 | - | - | - | - |
| Navaneet et al. | 27.16 | 0.808 | 0.228 | 50.30 | 23.47 | 0.840 | 0.188 | 27.97 | 29.75 | 0.903 | 0.247 | 42.77 | 24.63 | 0.823 | 0.239 | 104.3 |
| Trimming the fat | 27.13 | 0.798 | 0.248 | 20.057 | 23.68 | 0.831 | 0.210 | 8.555 | 29.42 | 0.897 | 0.267 | 12.49 | - | - | - | - |
| CompGS (high-rate) | 27.26 | 0.802 | 0.239 | 16.5 | 23.70 | 0.835 | 0.205 | 9.61 | 29.33 | 0.900 | 0.270 | 10.4 | - | - | - | - |
| CompGS (low-rate) | 26.79 | 0.791 | 0.258 | 11.0 | 23.105 | 0.815 | 0.235 | 5.89 | 28.99 | 0.900 | 0.280 | 7.00 | - | - | - | - |
| RDO-Gaussian (high-rate) | 27.05 | 0.802 | 0.239 | 23.46 | 23.34 | 0.835 | 0.195 | 12.02 | 29.63 | 0.902 | 0.252 | 18.00 | - | - | - | - |
| RDO-Gaussian (low-rate) | 24.43 | 0.683 | 0.406 | 1.71 | 22.09 | 0.755 | 0.318 | 1.32 | 28.38 | 0.872 | 0.331 | 1.22 | - | - | - | - |
| HAC (ECCV'24)(high-rate) | 27.77 | 0.811 | 0.230 | 21.87 | 24.40 | 0.853 | 0.177 | 11.24 | 30.34 | 0.906 | 0.258 | 6.35 | 27.05 | 0.868 | 0.217 | 26.16 |
| HAC (ECCV'24)(low-rate) | 26.11 | 0.759 | 0.312 | 5.96 | 23.11 | 0.809 | 0.238 | 3.68 | 28.62 | 0.888 | 0.302 | 2.64 | 24.56 | 0.768 | 0.327 | 9.74 |
| Ours (high-rate) | 27.77 | 0.809 | 0.241 | 12.35 | 24.41 | 0.853 | 0.189 | 6.93 | 30.29 | 0.909 | 0.269 | 3.56 | 27.35 | 0.886 | 0.183 | 26.59 |
| Ours (low-rate) | 25.82 | 0.730 | 0.362 | 1.72 | 22.97 | 0.786 | 0.293 | 1.42 | 28.53 | 0.878 | 0.336 | 0.93 | 25.19 | 0.808 | 0.279 | 10.14 |

Table 7: Results of Our approach for each scene from Mip-NeRF 360 dataset (Barron et al., 2022a).

| $\lambda_r$ | Scenes | PSNR↑ | SSIM↑ | LPIPS↓ | SIZE↓ |
|---|---|---|---|---|---|
| 0.002 | bicycle | 25.04 | 0.735 | 0.280 | 21.42 |
| | bonsai | 32.98 | 0.947 | 0.192 | 6.06 |
| | counter | 29.66 | 0.915 | 0.197 | 6.33 |
| | flower | 21.41 | 0.580 | 0.375 | 18.92 |
| | garden | 27.49 | 0.847 | 0.152 | 18.64 |
| | kitchen | 31.34 | 0.926 | 0.135 | 6.92 |
| | room | 31.95 | 0.924 | 0.212 | 4.11 |
| | stump | 26.79 | 0.766 | 0.271 | 11.23 |
| | treehill | 23.28 | 0.646 | 0.359 | 17.50 |
| | **AVG** | **27.77** | **0.809** | **0.241** | **12.35** |
| 0.004 | bicycle | 25.16 | 0.738 | 0.280 | 16.56 |
| | bonsai | 32.74 | 0.944 | 0.198 | 4.79 |
| | counter | 29.51 | 0.911 | 0.203 | 4.98 |
| | flower | 21.37 | 0.576 | 0.383 | 14.28 |
| | garden | 27.30 | 0.840 | 0.164 | 14.55 |
| | kitchen | 30.97 | 0.922 | 0.140 | 5.36 |
| | room | 31.73 | 0.921 | 0.221 | 3.22 |
| | stump | 26.87 | 0.766 | 0.278 | 8.50 |
| | treehill | 23.27 | 0.642 | 0.370 | 13.20 |
| | **AVG** | **27.66** | **0.807** | **0.249** | **9.49** |
| 0.01 | bicycle | 25.03 | 0.728 | 0.302 | 9.40 |
| | bonsai | 31.67 | 0.932 | 0.217 | 2.94 |
| | counter | 28.89 | 0.898 | 0.225 | 3.15 |
| | flower | 21.25 | 0.564 | 0.403 | 8.13 |
| | garden | 26.83 | 0.816 | 0.210 | 8.65 |
| | kitchen | 30.37 | 0.911 | 0.160 | 3.32 |
| | room | 31.24 | 0.911 | 0.243 | 2.06 |
| | stump | 26.62 | 0.752 | 0.309 | 4.45 |
| | treehill | 23.22 | 0.629 | 0.400 | 7.17 |
| | **AVG** | **27.24** | **0.793** | **0.274** | **5.47** |
| 0.015 | bicycle | 24.80 | 0.714 | 0.323 | 6.59 |
| | bonsai | 31.25 | 0.927 | 0.227 | 2.35 |
| | counter | 28.51 | 0.886 | 0.244 | 2.45 |
| | flower | 21.12 | 0.552 | 0.419 | 6.13 |
| | garden | 26.56 | 0.803 | 0.231 | 6.82 |
| | kitchen | 29.90 | 0.902 | 0.174 | 2.59 |
| | room | 30.90 | 0.902 | 0.262 | 1.60 |
| | stump | 26.38 | 0.736 | 0.333 | 3.25 |
| | treehill | 23.08 | 0.616 | 0.423 | 5.13 |
| | **AVG** | **26.94** | **0.782** | **0.293** | **4.10** |
| 0.03 | bicycle | 24.38 | 0.676 | 0.368 | 3.35 |
| | bonsai | 29.99 | 0.905 | 0.260 | 1.58 |
| | counter | 27.63 | 0.860 | 0.285 | 1.49 |
| | flower | 20.74 | 0.519 | 0.455 | 3.32 |
| | garden | 25.81 | 0.755 | 0.304 | 3.85 |
| | kitchen | 28.76 | 0.878 | 0.216 | 1.53 |
| | room | 30.19 | 0.887 | 0.292 | 1.05 |
| | stump | 25.64 | 0.690 | 0.389 | 1.83 |
| | treehill | 22.84 | 0.580 | 0.472 | 2.44 |
| | **AVG** | **26.22** | **0.750** | **0.338** | **2.27** |
| 0.04 | bicycle | 24.02 | 0.646 | 0.399 | 2.36 |
| | bonsai | 29.44 | 0.892 | 0.277 | 1.30 |
| | counter | 27.14 | 0.845 | 0.306 | 1.23 |
| | flower | 20.44 | 0.496 | 0.477 | 2.42 |
| | garden | 25.40 | 0.727 | 0.337 | 3.01 |
| | kitchen | 28.16 | 0.862 | 0.244 | 1.20 |
| | room | 29.73 | 0.879 | 0.307 | 0.88 |
| | stump | 25.30 | 0.666 | 0.415 | 1.35 |
| | treehill | 22.71 | 0.558 | 0.499 | 1.74 |
| | **AVG** | **25.82** | **0.730** | **0.362** | **1.72** |

Table 8: Results of Our approach for each scene from BungeeNeRF dataset (Xiangli et al., 2022).

| $\lambda_r$ | Scenes | PSNR↑ | SSIM↑ | LPIPS↓ | SIZE↓ |
|---|---|---|---|---|---|
| | amsterdam | 27.57 | 0.908 | 0.148 | 31.98 |
| | bilbao | 28.25 | 0.897 | 0.165 | 25.38 |
| | hollywood | 25.16 | 0.815 | 0.254 | 24.58 |
| 0.001 | pompidou | 25.82 | 0.864 | 0.214 | 28.83 |
| | quebec | 30.59 | 0.943 | 0.144 | 21.72 |
| | rome | 26.74 | 0.887 | 0.178 | 27.06 |
| | **AVG** | **27.35** | **0.886** | **0.184** | **26.59** |
| | amsterdam | 27.52 | 0.903 | 0.159 | 27.33 |
| | bilbao | 28.15 | 0.892 | 0.176 | 22.03 |
| | hollywood | 25.12 | 0.808 | 0.268 | 20.95 |
| 0.002 | pompidou | 25.70 | 0.857 | 0.224 | 24.26 |
| | quebec | 30.23 | 0.938 | 0.154 | 17.84 |
| | rome | 26.57 | 0.881 | 0.190 | 23.17 |
| | **AVG** | **27.22** | **0.880** | **0.195** | **22.60** |
| | amsterdam | 27.28 | 0.897 | 0.169 | 24.47 |
| | bilbao | 28.13 | 0.888 | 0.181 | 19.41 |
| | hollywood | 25.05 | 0.801 | 0.280 | 18.72 |
| 0.003 | pompidou | 25.69 | 0.854 | 0.232 | 21.52 |
| | quebec | 30.04 | 0.934 | 0.163 | 15.89 |
| | rome | 26.42 | 0.875 | 0.200 | 20.81 |
| | **AVG** | **27.10** | **0.875** | **0.204** | **20.14** |
| | amsterdam | 26.96 | 0.882 | 0.194 | 19.38 |
| | bilbao | 27.72 | 0.876 | 0.202 | 15.63 |
| | hollywood | 24.68 | 0.778 | 0.307 | 14.90 |
| 0.006 | pompidou | 25.21 | 0.839 | 0.251 | 17.22 |
| | quebec | 29.53 | 0.925 | 0.178 | 13.28 |
| | rome | 25.78 | 0.856 | 0.223 | 16.87 |
| | **AVG** | **26.65** | **0.859** | **0.226** | **16.21** |
| | amsterdam | 26.32 | 0.863 | 0.215 | 15.96 |
| | bilbao | 27.26 | 0.861 | 0.222 | 12.73 |
| | hollywood | 24.41 | 0.757 | 0.325 | 12.71 |
| 0.01 | pompidou | 24.86 | 0.825 | 0.267 | 14.33 |
| | quebec | 28.84 | 0.914 | 0.197 | 10.68 |
| | rome | 25.13 | 0.834 | 0.245 | 13.86 |
| | **AVG** | **26.14** | **0.842** | **0.245** | **13.38** |
| | amsterdam | 25.65 | 0.833 | 0.253 | 12.33 |
| | bilbao | 26.30 | 0.829 | 0.258 | 9.74 |
| | hollywood | 23.80 | 0.714 | 0.361 | 9.12 |
| 0.02 | pompidou | 23.78 | 0.789 | 0.298 | 10.75 |
| | quebec | 27.76 | 0.891 | 0.225 | 8.28 |
| | rome | 23.85 | 0.790 | 0.283 | 10.63 |
| | **AVG** | **25.19** | **0.808** | **0.280** | **10.14** |

Table 9: Results of Our approach for each scene from Tank & Temples dataset (Knapitsch et al., 2017).

| $\lambda_r$ | Scenes | PSNR↑ | SSIM↑ | LPIPS↓ | SIZE↓ |
|---|---|---|---|---|---|
| | train | 22.68 | 0.820 | 0.221 | 6.28 |
| 0.002 | truck | 26.14 | 0.885 | 0.157 | 7.57 |
| | **AVG** | **24.41** | **0.853** | **0.189** | **6.93** |
| | train | 22.45 | 0.817 | 0.226 | 5.06 |
| 0.004 | truck | 25.98 | 0.882 | 0.162 | 5.88 |
| | **AVG** | **24.22** | **0.850** | **0.194** | **5.47** |
| | train | 22.31 | 0.802 | 0.249 | 3.43 |
| 0.01 | truck | 25.68 | 0.871 | 0.184 | 3.74 |
| | **AVG** | **23.99** | **0.837** | **0.217** | **3.58** |
| | train | 22.23 | 0.794 | 0.262 | 2.71 |
| 0.015 | truck | 25.55 | 0.866 | 0.195 | 3.00 |
| | **AVG** | **23.89** | **0.830** | **0.228** | **2.86** |
| | train | 22.01 | 0.769 | 0.295 | 1.80 |
| 0.03 | truck | 24.97 | 0.842 | 0.238 | 1.89 |
| | **AVG** | **23.49** | **0.806** | **0.266** | **1.85** |
| | train | 21.42 | 0.750 | 0.315 | 1.47 |
| 0.04 | truck | 24.51 | 0.822 | 0.271 | 1.38 |
| | **AVG** | **22.97** | **0.786** | **0.293** | **1.42** |

Table 10: Results of Our approach for each scene from Deep Blending (Hedman et al., 2018).

| $\lambda_r$ | Scenes | PSNR↑ | SSIM↑ | LPIPS↓ | SIZE↓ |
|---|---|---|---|---|---|
| | drjohnson | 29.67 | 0.906 | 0.266 | 4.27 |
| 0.002 | playroom | 30.90 | 0.911 | 0.272 | 2.85 |
| | **AVG** | **30.29** | **0.909** | **0.269** | **3.56** |
| | drjohnson | 29.51 | 0.904 | 0.272 | 3.43 |
| 0.004 | playroom | 30.81 | 0.909 | 0.279 | 2.21 |
| | **AVG** | **30.16** | **0.906** | **0.275** | **2.82** |
| | drjohnson | 29.23 | 0.898 | 0.288 | 2.12 |
| 0.01 | playroom | 30.23 | 0.903 | 0.295 | 1.46 |
| | **AVG** | **29.73** | **0.900** | **0.292** | **1.79** |
| | drjohnson | 29.06 | 0.892 | 0.299 | 1.76 |
| 0.015 | playroom | 29.91 | 0.897 | 0.309 | 1.21 |
| | **AVG** | **29.48** | **0.894** | **0.304** | **1.48** |
| | drjohnson | 28.52 | 0.879 | 0.323 | 1.20 |
| 0.03 | playroom | 29.19 | 0.887 | 0.331 | 0.85 |
| | **AVG** | **28.85** | **0.883** | **0.327** | **1.03** |
| | drjohnson | 28.31 | 0.874 | 0.332 | 1.09 |
| 0.04 | playroom | 28.74 | 0.881 | 0.340 | 0.76 |
| | **AVG** | **28.53** | **0.878** | **0.336** | **0.93** |

Table 11: Results of HAC (Chen et al., 2024) for each scene from Mip-NeRF 360 dataset (Barron et al., 2022a).

| $\lambda_e$ | Scenes | PSNR↑ | SSIM↑ | LPIPS↓ | SIZE↓ |
|---|---|---|---|---|---|
| | bicycle | 25.11 | 0.742 | 0.259 | 39.15 |
| | bonsai | 32.97 | 0.948 | 0.180 | 12.72 |
| | counter | 29.74 | 0.918 | 0.184 | 10.44 |
| | flower | 21.27 | 0.575 | 0.377 | 27.55 |
| 0.001 | garden | 27.46 | 0.849 | 0.139 | 32.17 |
| | kitchen | 31.63 | 0.930 | 0.122 | 12.07 |
| | room | 31.90 | 0.926 | 0.198 | 7.85 |
| | stump | 26.59 | 0.763 | 0.264 | 25.26 |
| | treehill | 23.26 | 0.648 | 0.345 | 29.65 |
| | **AVG** | **27.77** | **0.811** | **0.230** | **21.87** |
| | bicycle | 25.10 | 0.742 | 0.262 | 33.14 |
| | bonsai | 32.70 | 0.945 | 0.184 | 10.51 |
| | counter | 29.65 | 0.915 | 0.189 | 8.88 |
| | flower | 21.32 | 0.576 | 0.377 | 23.73 |
| 0.002 | garden | 27.43 | 0.847 | 0.143 | 27.52 |
| | kitchen | 31.46 | 0.928 | 0.125 | 10.05 |
| | room | 31.87 | 0.925 | 0.201 | 6.47 |
| | stump | 26.59 | 0.761 | 0.268 | 21.75 |
| | treehill | 23.34 | 0.647 | 0.350 | 24.83 |
| | **AVG** | **27.72** | **0.809** | **0.233** | **18.54** |
| | bicycle | 25.05 | 0.742 | 0.264 | 27.54 |
| | bonsai | 32.28 | 0.942 | 0.189 | 8.56 |
| | counter | 29.35 | 0.911 | 0.195 | 7.26 |
| | flower | 21.26 | 0.572 | 0.381 | 19.59 |
| 0.004 | garden | 27.28 | 0.842 | 0.151 | 22.69 |
| | kitchen | 31.16 | 0.923 | 0.131 | 8.05 |
| | room | 31.55 | 0.921 | 0.208 | 5.53 |
| | stump | 26.58 | 0.762 | 0.269 | 18.11 |
| | treehill | 23.30 | 0.645 | 0.356 | 20.04 |
| | **AVG** | **27.53** | **0.807** | **0.238** | **15.26** |
| | bicycle | 24.79 | 0.733 | 0.284 | 17.80 |
| | bonsai | 31.27 | 0.933 | 0.208 | 5.16 |
| | counter | 28.68 | 0.898 | 0.220 | 4.54 |
| | flower | 21.18 | 0.561 | 0.400 | 12.15 |
| 0.01 | garden | 26.76 | 0.822 | 0.188 | 14.53 |
| | kitchen | 30.51 | 0.914 | 0.149 | 4.85 |
| | room | 31.20 | 0.912 | 0.234 | 3.27 |
| | stump | 26.54 | 0.752 | 0.296 | 11.21 |
| | treehill | 23.13 | 0.628 | 0.392 | 12.27 |
| | **AVG** | **27.12** | **0.795** | **0.263** | **9.53** |
| | bicycle | 24.60 | 0.717 | 0.306 | 14.97 |
| | bonsai | 30.51 | 0.922 | 0.225 | 4.24 |
| | counter | 27.78 | 0.878 | 0.248 | 3.30 |
| | flower | 20.84 | 0.537 | 0.425 | 8.90 |
| 0.02 | garden | 26.34 | 0.800 | 0.222 | 10.87 |
| | kitchen | 29.86 | 0.902 | 0.169 | 3.69 |
| | room | 30.51 | 0.900 | 0.256 | 2.56 |
| | stump | 26.20 | 0.730 | 0.326 | 8.68 |
| | treehill | 23.03 | 0.610 | 0.418 | 9.25 |
| | **AVG** | **26.63** | **0.777** | **0.288** | **7.38** |
| | bicycle | 24.16 | 0.694 | 0.333 | 11.59 |
| | bonsai | 29.63 | 0.909 | 0.242 | 3.72 |
| | counter | 27.19 | 0.864 | 0.269 | 2.68 |
| | flower | 20.57 | 0.513 | 0.449 | 7.17 |
| 0.035 | garden | 25.86 | 0.780 | 0.252 | 8.90 |
| | kitchen | 29.09 | 0.886 | 0.191 | 3.08 |
| | room | 29.94 | 0.889 | 0.276 | 2.15 |
| | stump | 25.77 | 0.707 | 0.358 | 7.03 |
| | treehill | 22.82 | 0.591 | 0.443 | 7.32 |
| | **AVG** | **26.11** | **0.759** | **0.312** | **5.96** |

Table 12: Results of HAC (Chen et al., 2024) for each scene from BungeeNeRF dataset (Xiangli et al., 2022).

| $\lambda_e$ | Scenes | PSNR↑ | SSIM↑ | LPIPS↓ | SIZE↓ |
|---|---|---|---|---|---|
| | amsterdam | 27.25 | 0.886 | 0.190 | 31.84 |
| | bilbao | 27.98 | 0.886 | 0.190 | 24.38 |
| | hollywood | 24.59 | 0.772 | 0.319 | 23.41 |
| 0.001 | pompidou | 25.58 | 0.851 | 0.236 | 29.19 |
| | quebec | 30.30 | 0.934 | 0.163 | 21.23 |
| | rome | 26.61 | 0.876 | 0.203 | 26.91 |
| | **AVG** | **27.05** | **0.868** | **0.217** | **26.16** |
| | amsterdam | 27.13 | 0.880 | 0.202 | 27.14 |
| | bilbao | 28.02 | 0.880 | 0.205 | 20.91 |
| | hollywood | 24.43 | 0.763 | 0.330 | 20.09 |
| 0.002 | pompidou | 25.27 | 0.842 | 0.249 | 24.85 |
| | quebec | 29.98 | 0.929 | 0.175 | 17.90 |
| | rome | 26.28 | 0.866 | 0.219 | 23.07 |
| | **AVG** | **26.85** | **0.860** | **0.230** | **22.33** |
| | amsterdam | 26.95 | 0.873 | 0.214 | 24.41 |
| | bilbao | 27.82 | 0.872 | 0.218 | 18.76 |
| | hollywood | 24.27 | 0.753 | 0.342 | 17.87 |
| 0.003 | pompidou | 25.34 | 0.837 | 0.255 | 22.49 |
| | quebec | 29.67 | 0.924 | 0.185 | 16.15 |
| | rome | 25.98 | 0.855 | 0.231 | 20.83 |
| | **AVG** | **26.67** | **0.852** | **0.241** | **20.08** |
| | amsterdam | 26.80 | 0.865 | 0.224 | 22.49 |
| | bilbao | 27.65 | 0.864 | 0.231 | 17.14 |
| | hollywood | 24.25 | 0.748 | 0.347 | 16.55 |
| 0.004 | pompidou | 25.16 | 0.829 | 0.266 | 20.40 |
| | quebec | 29.33 | 0.918 | 0.192 | 15.06 |
| | rome | 25.68 | 0.845 | 0.243 | 19.30 |
| | **AVG** | **26.48** | **0.845** | **0.250** | **18.49** |
| | amsterdam | 26.13 | 0.839 | 0.258 | 17.22 |
| | bilbao | 26.81 | 0.842 | 0.262 | 14.01 |
| | hollywood | 23.83 | 0.713 | 0.377 | 12.90 |
| 0.008 | pompidou | 24.48 | 0.807 | 0.289 | 16.10 |
| | quebec | 28.60 | 0.906 | 0.216 | 11.71 |
| | rome | 24.71 | 0.811 | 0.276 | 15.34 |
| | **AVG** | **25.76** | **0.820** | **0.280** | **14.55** |
| | amsterdam | 25.03 | 0.788 | 0.308 | 11.92 |
| | bilbao | 25.84 | 0.799 | 0.308 | 9.08 |
| | hollywood | 22.92 | 0.640 | 0.425 | 8.38 |
| 0.02 | pompidou | 23.16 | 0.759 | 0.332 | 10.43 |
| | quebec | 27.15 | 0.872 | 0.262 | 8.18 |
| | rome | 23.25 | 0.750 | 0.327 | 10.47 |
| | **AVG** | **24.56** | **0.768** | **0.327** | **9.74** |

Table 13: Results of HAC (Chen et al., 2024) for each scene from Tank & Temples dataset (Knapitsch et al., 2017).

| $\lambda_e$ | Scenes | PSNR↑ | SSIM↑ | LPIPS↓ | SIZE↓ |
|---|---|---|---|---|---|
| | truck | 26.02 | 0.883 | 0.147 | 12.42 |
| 0.001 | train | 22.78 | 0.823 | 0.207 | 10.07 |
| | **AVG** | **24.40** | **0.853** | **0.177** | **11.24** |
| | truck | 25.88 | 0.878 | 0.158 | 9.26 |
| 0.004 | train | 22.19 | 0.815 | 0.216 | 6.94 |
| | **AVG** | **24.04** | **0.846** | **0.187** | **8.10** |
| | truck | 25.69 | 0.874 | 0.172 | 7.04 |
| 0.01 | train | 22.13 | 0.807 | 0.230 | 5.26 |
| | **AVG** | **23.91** | **0.841** | **0.201** | **6.15** |
| | truck | 25.36 | 0.863 | 0.189 | 5.59 |
| 0.02 | train | 22.02 | 0.795 | 0.250 | 3.97 |
| | **AVG** | **23.69** | **0.829** | **0.219** | **4.78** |
| | truck | 24.96 | 0.853 | 0.208 | 5.04 |
| 0.035 | train | 21.96 | 0.783 | 0.266 | 3.17 |
| | **AVG** | **23.46** | **0.818** | **0.237** | **4.10** |
| | truck | 24.81 | 0.847 | 0.216 | 4.64 |
| 0.045 | train | 21.42 | 0.771 | 0.279 | 2.72 |
| | **AVG** | **23.11** | **0.809** | **0.248** | **3.68** |

Table 14: Results of HAC (Chen et al., 2024) for each scene from Deep Blending dataset (Hedman et al., 2018).

| $\lambda_e$ | Scenes | PSNR↑ | SSIM↑ | LPIPS↓ | SIZE↓ |
|---|---|---|---|---|---|
| | playroom | 30.84 | 0.906 | 0.262 | 5.03 |
| 0.001 | drjohnson | 29.85 | 0.906 | 0.255 | 7.67 |
| | **AVG** | **30.34** | **0.906** | **0.258** | **6.35** |
| | playroom | 30.66 | 0.905 | 0.265 | 4.12 |
| 0.002 | drjohnson | 29.69 | 0.905 | 0.258 | 6.51 |
| | **AVG** | **30.17** | **0.905** | **0.262** | **5.32** |
| | playroom | 30.44 | 0.902 | 0.272 | 3.15 |
| 0.004 | drjohnson | 29.53 | 0.903 | 0.265 | 5.55 |
| | **AVG** | **29.98** | **0.902** | **0.269** | **4.35** |
| | playroom | 30.31 | 0.906 | 0.279 | 2.58 |
| 0.01 | drjohnson | 29.34 | 0.901 | 0.275 | 4.23 |
| | **AVG** | **29.83** | **0.903** | **0.277** | **3.40** |
| | playroom | 29.87 | 0.900 | 0.292 | 2.23 |
| 0.02 | drjohnson | 28.81 | 0.893 | 0.288 | 3.65 |
| | **AVG** | **29.34** | **0.897** | **0.290** | **2.94** |
| | playroom | 29.26 | 0.893 | 0.302 | 2.01 |
| 0.035 | drjohnson | 27.97 | 0.883 | 0.302 | 3.26 |
| | **AVG** | **28.62** | **0.888** | **0.302** | **2.64** |

Table 15: Results of ScaffoldGS (Lu et al., 2024) for all evaluated datasets.

| Datasets | Scenes | PSNR↑ | SSIM↑ | LPIPS↓ | SIZE↓ |
|---|---|---|---|---|---|
| Mip-NeRF360 | bicycle | 24.50 | 0.705 | 0.306 | 248.00 |
| | bonsai | 32.70 | 0.946 | 0.185 | 258.00 |
| | counter | 29.34 | 0.914 | 0.191 | 194.00 |
| | flower | 21.14 | 0.566 | 0.417 | 253.00 |
| | garden | 27.17 | 0.842 | 0.146 | 271.00 |
| | kitchen | 31.30 | 0.928 | 0.126 | 173.00 |
| | room | 31.93 | 0.925 | 0.202 | 133.00 |
| | stump | 26.27 | 0.784 | 0.284 | 493.00 |
| | treehill | 23.19 | 0.642 | 0.410 | 262.00 |
| | **AVG** | **27.50** | **0.806** | **0.252** | **253.89** |
| Tank&Temples | truck | 25.77 | 0.883 | 0.147 | 107.00 |
| | train | 22.15 | 0.822 | 0.206 | 66.00 |
| | **AVG** | **23.96** | **0.853** | **0.177** | **86.50** |
| DeepBlending | playroom | 30.62 | 0.904 | 0.258 | 63.00 |
| | drjohnson | 29.80 | 0.907 | 0.250 | 69.00 |
| | **AVG** | **30.21** | **0.906** | **0.254** | **66.00** |
| BungeeNeRF | amsterdam | 27.16 | 0.898 | 0.188 | 223.00 |
| | bilbao | 26.60 | 0.857 | 0.257 | 178.00 |
| | hollywood | 24.49 | 0.787 | 0.318 | 155.00 |
| | pompidou | 24.94 | 0.839 | 0.271 | 209.00 |
| | quebec | 30.28 | 0.936 | 0.190 | 159.00 |
| | rome | 26.23 | 0.873 | 0.225 | 174.00 |
| | **AVG** | **26.62** | **0.865** | **0.241** | **183.00** |

