# OpenReview forum: "CAT-3DGS: A Context-Adaptive Triplane Approach to Rate-Distortion-Optimized 3DGS Compression"
_ICLR.cc/2025/Conference — ICLR 2025 Poster_

### Official Review · Reviewer_VHE1 · 2024-11-03

**Soundness:** 3
**Presentation:** 2
**Contribution:** 2
**Rating:** 6
**Confidence:** 4

**Summary:**

In this paper, the authors present a context-adaptive and rate-distortion-optimized 3DGS coding approach named CAT-3DGS. The approach introduces several modules for efficient attribute 3DGS compression, including a triplane-based hyperprior, spatial autoregressive models (SARM), channel-wise autoregressive models (CARM) and view frequency-aware masking. These modules are jointly trained in a rate-distortion-optimized framework. Experimental results demonstrate that CAT-3DGS achieves rate reductions of 78× and 26× compared to 3DGS and Scaffold-GS, respectively, while surpassing existing 3DGS compression schemes in rate-distortion performance.

**Strengths:**

1.The coding architecture is well-structured and theoretically solid, with four distinct modules specifically designed to address various redundancies.

2.The proposed method achieves exceptional rate-distortion performance, surpassing other 3DGS compression techniques.

**Weaknesses:**

1.The effectiveness of view frequency-aware masking is somewhat ambiguous. In Sec. 5.3, the authors report a 16% BD-rate reduction due to view frequency-aware masking. However, it’s unclear whether the anchor configuration (labeled as “w/o VFM” in Fig. 10) excludes masking altogether, or retains the masking operation but removes the weights p_{n,k} only.

2.Spatial redundancies within attributes are not fully exploited. The proposed SARM only addresses inter-redundancies within triplanes (hyperpriors) rather than anchors. Given the fact that spatial redundancies exist between anchors [1], this aspect is not leveraged in the current framework.

[1] Yufei Wang, Zhihao Li, Lanqing Guo, Wenhan Yang, Alex C Kot, and Bihan Wen. Contextgs: Compact 3d gaussian splatting with anchor level context model. arXiv preprint arXiv:2405.20721, 2024b.

**Questions:**

Please see weakness

---

> ### Author Response · Authors · 2024-11-20
> **Response to Reviewer VHE1**
>
> >  **[W1] The effectiveness of view frequency-aware masking is somewhat ambiguous. It’s unclear whether the anchor configuration (labeled as “w/o VFM” in Fig. 10) excludes masking altogether, or retains the masking operation but removes the weights $p_{n,k}$ only.**
>
> Thank you for pointing out this ambiguity. The curve labeled as 'w/o VFM' in Figure 10 still **retains the masking operation but removes the weights $p_{n,k}$ only**. In the revised manuscript, we have updated the text in lines 514-515 and 743-744 accordingly. Please see the text in red color.
>
> ---
>
> >  **[W2] Spatial redundancies within attributes are not fully exploited. The proposed SARM only addresses inter-redundancies within triplanes (hyperpriors) rather than anchors.**
>
> This is an interesting point! In Figure 4 of our revised manuscript, we provide additional rate-distortion plots, including ContextGS, which directly leverages the decoded anchors' attributes as context for decoding the remaining anchors' attributes. We see that our CAT-3DGS outperforms ContextGS on most of the datasets. This suggests that **our triplane-based hyperprior and SARM are effective in capturing the inter-redundancies among the anchors' attributes**. We remark that ContextGS `[R1]` and our method could be combined to further enhance the coding efficiency.
>
> ---
>
> **We thank the reviewer for their insightful and constructive comments.** We have carefully addressed all the concerns and comments and hope the reviewer find our responses satisfactory. If so, we would greatly appreciated if the reviewer could consider reflecting this in the rating for this early attempt at efficient coding of 3DGS representations, which we believe would invite more contributions from the community to advance this largely unexplored yet important area.
>
> ---
>
> ***References***
>
> `[R1]` Y. Wang et al., "Contextgs: Compact 3d gaussian splatting with anchor level context model," arXiv, 2024.

---

> > ### Author Response · Authors · 2024-12-01
> >
> > Dear Reviewer VHE1, the interactive rebuttal period is going to end soon. We have addressed your editorial comment [W1] in our earlier response. To tackle your second concern [W2] (i.e. spatial redundancies within attributes are not fully exploited), we have additionally compared our CAT-3DGS with ContextGS, which performs contextual entropy coding directly on attributes. The results in Fig. 4 of our revised manuscript indicate that our CAT-3DGS performs better than or comparably to ContextGS. Note that ContextGS is a concurrent work at the time of submission and is currently only available on arXiv. We note that our triplane-based hyperprior and context coding can be combined with the idea from ContextGS. This is among our future work. Last but not least, we would be happy to discuss any further comment you may have. If you find our responses satisfactory, we would appreciate your kind consideration of increasing your rating.

---

> ### Author Response · Authors · 2024-11-25
> **Response to Reviewer VHE1**
>
> Dear Reviewer, we thank you very much for the effort you put into reviewing our paper. We have addressed your comments to the extent possible. As **the interactive discussion phase for the rebuttal will end on Nov. 27 AOE**, we look very much forward to your further comments (if any). If you find our responses satisfactory, we would ask your kind consideration for increasing your rating on this early and novel attempt at addressing 3DGS coding.

---

### Official Review · Reviewer_GetA · 2024-11-03

**Soundness:** 3
**Presentation:** 3
**Contribution:** 2
**Rating:** 6
**Confidence:** 4

**Summary:**

The paper introduces CAT-3DGS, a context-adaptive compression method for 3D Gaussian splatting (3DGS) that optimizes rate-distortion performance. CAT-3DGS uses multi-scale triplanes to embed spatial correlations of Gaussian primitives and employs autoregressive models for encoding. Additionally, it uses a channel-wise autoregressive model to enhance compression efficiency within each primitive and introduces a view frequency-aware masking mechanism to skip encoding primitives that minimally impact image quality. Experimental results demonstrate the effectiveness of the proposed method.

**Strengths:**

1. PCA-guided triplanes as the hyperprior
2. Channel-wise Autoregressive Model: Enhances compression by leveraging internal correlations within each voxel.
3. View Frequency-aware Masking Mechanism: Skips encoding for primitives with minimal impact on rendering quality, reducing redundancy and enhancing overall performance.

**Weaknesses:**

1. The ablation study of PCA guidance need to be included since there are already serval works that utilizes triplanes to compress the 3DGS/Nerf.
2. The efficiency of the model. The model seems a bit complicated. The coding time greatly increased due to the proposed context model. Besides, how about the training time?
3. Missing quantitative comparison with ContextGS, which is an important baseline method which also uses context model. For many benchmarks, the improvements appear somewhat marginal.
4. Figure 2 may not be entirely accurate; to my knowledge, CompGS does not utilize a context model.

**Questions:**

Please see the weakness part.

---

> ### Author Response · Authors · 2024-11-20
> **Response to Reviewer GetA (1/2)**
>
> >  **[W1] The ablation study of PCA guidance need to be included since there are already serval works that utilizes triplanes to compress the 3DGS/Nerf.**
>
> First, to our best knowledge, most of the prior works that utilize triplanes for 3DGS/NeRF scene representations do NOT consider the need to encode these representations into compressed bistreams for transmission, which requires entropy coding and rate-distortion optimized training. Our CAT-3DGS distinguishes from these works by focusing specifically on how to entropy encode these scene representations in a rate-distortion optimized fashion. Notably, it involves triplanes as the hyperprior.
>
> Second, as per the request, we follow the ablation setup in the paper to report the BD-rate comparison between CAT-3DGS with and without PCA on Mip-NeRF 360 dataset (9 scenes). The variant without PCA serves as the anchor for BD-rate evaluation. The negative BD-rate number of the variant with PCA shows that **PCA brings a marginal rate saving (2.2\%)**. In the paper, we **consider PCA to be a pre-processing step, not our major contribution**.
>
> | Method                | CAT-3DGS w/o PCA | CAT-3DGS (w/ PCA) |
> |-----------------------|------------------|-------------------|
> | **BD-rate (%)**       | 0               | -2.2             |
>
> ---
>
> >  **[W2] The efficiency of the model. The model seems a bit complicated. The coding time greatly increased due to the proposed context model. Besides, how about the training time?**
>
> Thank you for noting this. In reporting the decoding runtime of our CAT-3DGS in the initial submission, the runtime spent on the rate estimation for coding triplanes was accidentally included, leading to our prolonged decoding time. We have updated our decoding time in Table 2 of the revised manuscript and provide the following table, which includes HAC, as well as the initial and revised versions of CAT-3DGS.
> | **Scene**      | **Method**         | **Triplane (s)** | **Anchor Attributes (s)** | **Total (s)** |
> |-----------------|--------------------|------------------|---------------------------|---------------|
> | **room**        | HAC                | N/A              | -                         | 6.6           |
> |                 | CAT-3DGS           | 11.4             | 2.3                       | 13.7          |
> |                 | CAT-3DGS (revised) | 11.4             | 2.2                       | 13.6          |
> | **amsterdam**   | HAC                | N/A              | -                         | 11.3          |
> |                 | CAT-3DGS           | 47.4             | 67.7                      | 115.0         |
> |                 | CAT-3DGS (revised) | 47.4             | 17.0                      | 64.4          |
>
> As per the request, we provide the training time comparison as follows. The training time of CAT-3DGS is on par with or higher than that of HAC. The longer training time may be attributed to the differences in the **training recipe** and **the use of SARM in estimating the triplane's bit rate**. Additionally, for the Amsterdam scene, CAT-3DGS requires approximately twice the training time of HAC. This is due primarily to the larger number of anchor points (1599K anchors), which significantly **increases the triplane query time**. In contrast, the room scene has relatively fewer anchor points (109K anchors), resulting in a less noticeable increase in training time. Nonetheless, we stress that the training time is highly variable and dependent on the training recipe. It is among our future work to speed up the training process.
>
> | Scene      | Method     | $\lambda_r$    | Training Time (min) |
> |------------|------------|--------|---------------------|
> | **room**   | HAC        | 0.35   | 42                  |
> |            | CAT-3DGS   | 0.4    | 53                  |
> | **amsterdam** | HAC     | 0.35   | 53                  |
> |            | CAT-3DGS   | 0.4    | 125                 |
>
> ---
>
> >  **[W3] Missing quantitative comparison with ContextGS.**
>
> We did not compare our scheme with ContextGS for a number of reasons. First, the code at their GitHub page (https://github.com/wyf0912/ContextGS) is regrettably NOT available yet. Second, the authors reported only two rate-distortion points in their paper, hindering a comprehensive rate-distortion comparison with our and the other existing approaches. Lastly, we notice the ICLR FAQ for Reviewers (https://iclr.cc/Conferences/2025/ReviewerGuide) that "if a paper was published (i.e., at a peer-reviewed venue) on or after July 1, 2024, authors are not required to compare their own work to that paper." At the time of submission, ContextGS was still an arXiv preprint (dated October 1, 2024).
>
> Nonetheless, as per the request, we provide **additional rate-distortion plots, including ContextGS, in Figure 4 of our revised manuscript**. Note that the results of ContextGS are from their paper. Roughly speaking, our CAT-3DGS performs better or comparably to ContextGS.

---

> ### Author Response · Authors · 2024-11-20
> **Response to Reviewer GetA (2/2)**
>
> >  **[W4] Figure 2 may not be entirely accurate; to my knowledge, CompGS does not utilize a context model.**
>
> We clarify that CompGS has its own 3DGS representation (which is different from ScaffoldGS). It includes the anchor and coupled primitives. **It does utilize a context model although not autoregressive-based.** As stated in Section 3.3 of their paper `[R1]`, "*Furthermore, $\tilde{f}{\omega}$ is used as contexts to model the probability distributions of $\tilde{\Sigma}{\omega}$ and $\tilde{g}_{k}$.*" In their work, the decoded reference embeddings of the anchor primitives serve as the contextual information for coding both the covariance of the anchor primitives and the residual embeddings of the coupled primitives.
>
> ---
>
>
> **We thank the reviewer for their insightful and constructive comments.** We have carefully addressed all the concerns and comments and hope the reviewer find our responses satisfactory. If so, we would greatly appreciated if the reviewer could consider reflecting this in the rating for this early attempt at efficient coding of 3DGS representations, which we believe would invite more contributions from the community to advance this largely unexplored yet important area.
>
> ---
>
> ***References***
>
> `[R1]` X. Liu et al., "Compgs: Efficient 3d scene representation via compressed gaussian splatting," arXiv, 2024.

---

> > ### Comment · Reviewer_GetA · 2024-11-25
> >
> > Thank you for your reply.
> >
> > Strictly speaking (at least from the perspective of deep learning-based compression, where the context model specifically refers to autoregressive models), if the criterion for determining whether it is a "context model" is based on "has its own 3DGS representation" then HAC can also be considered a "context model".
> >
> > I think it is necessary to distinguish between an autoregressive model (context model) and owning a 3DGS representation (usually referred to as a hyper-prior).

---

> > > ### Author Response · Authors · 2024-11-25
> > > **Response to Reviewer GetA**
> > >
> > > **Thank you for your valuable feedback, which has greatly helped us improve the clarity and accuracy of our paper.** We have followed your suggestion and refer the context model specifically to the autoregressive (AR) model, be it the spatial or channel-wise AR model in the latent or other domain. As such, we consider **CompGS to be in the same category of HAC**. In CompGS, there are two types of primitives: anchor and coupled primitives. The reference embedding of the anchor primitive plays a similar role to the hyperprior, serving to decode the attributes of the coupled primitives (which act more like the main latents in learned image compression) and the covariance of the anchor primitive itself. We have updated Figure 2 and the text in lines 153-155 in the revised manuscript to reflect these changes.

---

> > > > ### Comment · Reviewer_GetA · 2024-11-26
> > > >
> > > > Thank you for your response. Most of my concerns have been resolved, and I have updated my score to 6.

---

> > > > > ### Author Response · Authors · 2024-11-26
> > > > > **Response to Reviewer GetA**
> > > > >
> > > > > Thank you for your favorable rating and positive feedback.

---

> ### Author Response · Authors · 2024-11-25
> **Response to Reviewer GetA**
>
> Dear Reviewer, we thank you very much for the effort you put into reviewing our paper. We have addressed your comments to the extent possible. As **the interactive discussion phase for the rebuttal will end on Nov. 27 AOE**, we look very much forward to your further comments (if any). If you find our responses satisfactory, we would ask your kind consideration for increasing your rating on this early and novel attempt at addressing 3DGS coding.

---

### Official Review · Reviewer_tuJM · 2024-11-03

**Soundness:** 3
**Presentation:** 3
**Contribution:** 2
**Rating:** 6
**Confidence:** 5

**Summary:**

The paper presents CAT-3DGS, a context-adaptive triplane approach for compressing 3D Gaussian Splatting (3DGS) data with a focus on optimizing the rate-distortion trade-off. CAT-3DGS introduces multi-scale triplanes aligned with the principal axes of Gaussian primitives in 3D space, leveraging both spatial and channel-wise autoregressive models to capture inter and intra correlations for efficient compression. This approach also includes a view frequency-aware masking mechanism to skip encoding less impactful Gaussian primitives. Experiments demonstrate CAT-3DGS achieves superior rate-distortion performance compared to prior 3DGS compression models like HAC, particularly on commonly used real-world datasets.

**Strengths:**

1. Innovative Triplane-Based Hyperprior:

By projecting 3D Gaussian primitives onto triplanes, the paper introduces an efficient hyperprior structure that captures spatial correlation, which improves entropy coding performance.

2. Enhanced Coding Efficiency:

The combination of spatial and channel-wise autoregressive models effectively utilizes both inter and intra correlations, which is particularly valuable in reducing the bit rate while maintaining high rendering quality.

3. Practical Masking Mechanism:

The view frequency-aware masking mechanism selectively encodes only those primitives crucial for rendering, further optimizing storage and bandwidth usage.

4. Comprehensive Experimental Evaluation:

The method is tested across several datasets and compared against notable baselines, showcasing significant rate savings and quality retention.

**Weaknesses:**

1. (Minor) Autoregressive Model Use:

While the channel-wise and pixel-wise autoregressive models are novel in this context, they are a commonly applied strategy in image compression tasks. However, this paper does make a unique contribution by applying it to 3DGS.

2. Lack of Comparative Analysis with ContextGS:

Although ContextGS is cited as a related work, there is no direct performance comparison with it. Given that ContextGS provides detailed results and its code is available, a comparative evaluation would enhance clarity on CAT-3DGS's improvements over previous models, especially regarding contextual coding efficiency.​

**Questions:**

I do not have other questions.

---

> ### Author Response · Authors · 2024-11-20
> **Response to Reviewer tuJM**
>
> > **[W1] Autoregressive Model Use: While the channel-wise and pixel-wise autoregressive models are novel in this context, they are a commonly applied strategy in image compression tasks. However, this paper does make a unique contribution by applying it to 3DGS.**
>
> We thank the reviewer for recognizing our autoregressive models as being novel in the present context. We stress that this work marks the first attempt to introduce autoregressive models to the triplane-based hyperprior for efficient coding of the 3DGS representation. These techniques along with the view-frequency aware masking achieve the state-of-the-art coding performance. We believe this work will inspire more contributions from the community to advance this largely unexplored yet fast progressing area.
>
> ---
>
> > **[W2] Lack of comparative analysis with ContextGS.**
>
> We did not compare our scheme with ContextGS for a number of reasons. First, the code at their GitHub page (https://github.com/wyf0912/ContextGS) is regrettably NOT available yet. Second, the authors reported only two rate-distortion points in their paper, hindering a comprehensive rate-distortion comparison with our and the other existing approaches. Lastly, we notice the ICLR FAQ for Reviewers (https://iclr.cc/Conferences/2025/ReviewerGuide) that "if a paper was published (i.e., at a peer-reviewed venue) on or after July 1, 2024, authors are not required to compare their own work to that paper." At the time of submission, ContextGS was still an arXiv preprint (dated October 1, 2024).
>
> Nonetheless, as per the request, we **provide additional rate-distortion plots, including ContextGS, in Figure 4 of our revised manuscript**. Note that the results of ContextGS are from their paper. Roughly speaking, our CAT-3DGS performs better or comparably to ContextGS.
>
> ---
>
> **We thank the reviewer for their insightful and constructive comments.** We have carefully addressed all the concerns and comments and hope the reviewer find our responses satisfactory. If so, we would greatly appreciated if the reviewer could consider reflecting this in the rating for this early attempt at efficient coding of 3DGS representations, which we believe would invite more contributions from the community to advance this largely unexplored yet important area.

---

> ### Author Response · Authors · 2024-11-25
> **Response to Reviewer tuJM**
>
> Dear Reviewer, we thank you very much for the effort you put into reviewing our paper. We have addressed your comments to the extent possible. As **the interactive discussion phase for the rebuttal will end on Nov. 27 AOE**, we look very much forward to your further comments (if any). If you find our responses satisfactory, we would ask your kind consideration for increasing your rating on this early and novel attempt at addressing 3DGS coding.

---

> > ### Comment · Reviewer_tuJM · 2024-11-26
> >
> > Thank you for the detailed rebuttal and additional clarifications. While I appreciate the effort and see the value in your contributions, I believe the current score accurately reflects the balance between novelty and impact.

---

> > > ### Author Response · Authors · 2024-11-26
> > >
> > > Thank you for carefully reviewing our paper and for your positive feedback.

---

### Official Review · Reviewer_stVt · 2024-11-04

**Soundness:** 4
**Presentation:** 3
**Contribution:** 3
**Rating:** 6
**Confidence:** 4

**Summary:**

This paper proposed a context-adaptive triplane based hyperprior entropy model to capture the inter correlation among Gaussian primitives in the 3D space  (i.e. spatial correlation) for spatial autoregressive coding in the projected 2D planes. The channel-wise autoregressive coding is performed to leverage the intra correlation within each individual Gaussian primitive. Besides, the view frequency-aware masking mechanism is proposed to actively skip from coding those Gaussian primitives with little impact on the rendering quality. Experimental results show that the proposed CAT-3DGS achieves the state-of-the-art compression performance on the commonly used real-world datasets.

**Strengths:**

- ***Novelty of Framework:*** The key idea of decomposing the dense 3D hash-graid in HAC with triplane hyperprior entropy model is novel and reasonable.
- ***Technological Innovation:*** This paper proposed the SARM and FARM modules to exploit the inter-correlation and intra-correlation, respectively. Speciflly, the view frequency-aware masking mechanism is designed to to skip less critical Gaussian primitives from coding.
- This paper is well organized and easy to follow.

**Weaknesses:**

- Ablation experiment for CARM is not sufficient, the proposed method is only evaluated on one scenario, *e.g.*, bicycle scene.
- The paper is based on the work of HAC. However, the rate parameter setting used by HAC in the comparison experiment is not the same as that used by the proposed method. I doubt the fairness of this setting.

**Questions:**

- The FARM proposed by the author is interesting. According to the manuscript, the anchor feature is a latent representation of the anchor point, which does not have a clear mathematical correlation like the spherical correlation coefficient of 3DGS. What is the motivation of the proposed channel-wise autoregressive processing of anchor features?
- It is noted that the rate parameters setting used by the authors on different datasets are not consistent. Why?

---

> ### Author Response · Authors · 2024-11-20
> **Response to Reviewer stVt (1/2)**
>
> > **[W1] Ablation experiment for CARM is not sufficient, the proposed method is only evaluated on one scenario, e.g., bicycle scene.**
>
> **Table 1 of our initial submission presents the ablation results on CARM based on all the scenes (9 in total) in the Mip-NeRF 360 dataset**. Our unevenly partitioned CARM achieves an 11.9\% BD-rate saving compared to the variant without CARM. We shall make this clear in the final paper. Due to the page limit, Figure 7 provides a breakdown analysis for the bicycle scene only. As per the request, we provide **additional breakdown results for *flower* scene and *bonsai* scene in Figure 11 in Appendix F of our revised manuscript**. Consistent with the results presented in our paper, adopting CARM efficiently reduces the compressed size of the latent features.
>
> ---
>
> > **[W2] The paper is based on the work of HAC. However, the rate parameter setting used by HAC in the comparison experiment is not the same as that used by the proposed method. I doubt the fairness of this setting.**
>
> First, we clarify that we adopt the same ScaffoldGS representation as HAC; however, **our CAT-3DGS is NOT based on HAC**. In fact, our CAT-3DGS differs from HAC in several significant ways: (1) it utilizes triplanes as the hyperprior instead of the hash grids in HAC, (2) it introduces SARM and CARM for contextual entropy coding, which is NOT seen in HAC, and (3) it incorporates a view frequency-aware masking mechanism to filter out unimportant anchors. These novel elements are first proposed in this work.
>
> Second, since our CAT-3DGS is fundamentally different from HAC, it is natural that our rate parameters differ from those of HAC. To ensure a fair comparison and align with the common practice adopted by the learned image/video coding community, **we choose the rate parameters for both HAC and our CAT-3DGS in such a way that their reconstructed images span a wide (and approximately the same) range of quality to facilitate BD-rate evaluation**. Following this common practice approach, which is widely adopted by the image/video coding community, **our CAT-3DGS consistently achieves superior BD-rate results across various datasets**, as shown in the following table.
>
> | Dataset           | HAC (Anchor) | Our CAT-3DGS |
> |-------------------|--------------|--------------|
> | Mip-NeRF 360      | 0%           | -52.1%       |
> | BungeeNeRF        | 0%           | -19.8%       |
> | Tanks & Temples   | 0%           | -49.9%       |
> | Deep Blending     | 0%           | -50.6%       |
>
> [**Note**: The BD-rate is a common rate-distortion metric that quantifies the average bitrate saving achieved by a tested codec (the red curve at https://imgur.com/a/eRv2aWN) when compared with an anchor codec (the blue curve) at the same PSNR levels. A negative BD-rate value in percentage terms (e.g. -50\%) indicates that when compared with the anchor codec, the tested codec on average achieves a 50\% rate saving at the same quality level.]
>
> ---
>
> > **[Q1] The FARM proposed by the author is interesting. According to the manuscript, the anchor feature is a latent representation of the anchor point, which does not have a clear mathematical correlation like the spherical correlation coefficient of 3DGS. What is the motivation of the proposed channel-wise autoregressive processing of anchor features?**
>
> The anchor features are learnable latent parameters and encode collectively an anchor's attributes. During training, there is no constraint imposed to ensure that the anchor features are de-correlated. Our CARM is thus proposed to **leverage the potential correlation among the anchor features for better coding efficiency**. Our experimental results confirm the effectiveness of CARM, corroborating the intra-correlation among the anchor features.

---

> ### Author Response · Authors · 2024-11-20
> **Response to Reviewer stVt (2/2)**
>
> > **[Q2] It is noted that the rate parameters setting used by the authors on different datasets are not consistent. Why?**
>
> Following the common practice adopted by the learned image/video coding community, we aim to characterize better the rate-distortion performance of the competing methods over a wide range of quality levels and bitrates. We thus **extend the rate parameters of HAC reported in their paper to include more rate-distortion points**. We then **repeat the same process for our CAT-3DGS**. Efforts are made to align its quality levels with those of HAC (the major anchor). In doing so, it happens that the same rate parameters are chosen for CAT-3DGS on Mip-NeRF 360, Deep Blending, and Tank \& Temple datasets, while **slightly different rate parameters are used for BungeeNeRF dataset**. **We stress that this does not induce any bias towards their rate-distortion performance**.
>
> ---
>
> **We thank the reviewer for their insightful and constructive comments.** We have carefully addressed all the concerns and comments and hope the reviewer find our responses satisfactory. If so, we would greatly appreciated if the reviewer could consider reflecting this in the rating for this early attempt at efficient coding of 3DGS representations, which we believe would invite more contributions from the community to advance this largely unexplored yet important area.

---

> ### Author Response · Authors · 2024-11-25
> **Response to Reviewer stVt**
>
> Dear Reviewer, we thank you very much for the effort you put into reviewing our paper. We have addressed your comments to the extent possible. As the **interactive discussion phase for the rebuttal will end on Nov. 27 AOE**, we look very much forward to your further comments (if any).  If you find our responses satisfactory, we would ask your kind consideration for increasing your rating on this early and novel attempt at addressing 3DGS coding.

---

### Official Review · Reviewer_egdp · 2024-11-04

**Soundness:** 3
**Presentation:** 3
**Contribution:** 3
**Rating:** 6
**Confidence:** 4

**Summary:**

The paper introduces a new method called CAT-3DGS for compressing 3D Gaussian Splatting (3DGS) representations. This method aims to optimize the rate-distortion trade-off by using a context-adaptive triplane approach. It captures spatial correlations through multi-scale triplanes and leverages intra correlations within Gaussian primitives for efficient coding. Additionally, it incorporates a view frequency-aware masking mechanism to skip less impactful primitives, achieving state-of-the-art compression performance on real-world datasets.

**Strengths:**

+ The paper introduces a new compression method for 3DGS that utilizes a triplane-based hyperprior. This method leverages multi-scale triplanes oriented along the principal axes of Gaussian primitives.
+ The paper presents a novel pruning approach in 3DGS using a view frequency-aware masking mechanism. This mechanism assesses the significance of Gaussian primitives based on their impact on rendering quality, allowing less critical ones to be skipped during coding.
+ The paper presents comprehensive ablation studies of various modules in the proposed compression method.

**Weaknesses:**

- Entropy coding is a variable-length coding technique, typically decoded sequentially. However, 3DGC, used in rendering, is a highly parallelized method. The paper does not clearly explain how the proposed compression method can manage this parallelization without impacting rendering speed.
- The paper does not address decoding complexity. While CHARM enhances compression performance, it could significantly impact decoding speed, which in turn may affect rendering speed.
- The paper does not clarify whether real entropy coding was used during rendering or if the bit-rate was calculated based on theoretical entropy coding.

**Questions:**

* Are there scenarios where CAT-3DGS underperforms? Which modules in CAT-3DGS contribute to the drop in performance?

---

> ### Author Response · Authors · 2024-11-20
> **Response to Reviewer egdp**
>
> > **[W1] Entropy coding is a variable-length coding technique, typically decoded sequentially. However, 3DGC, used in rendering, is a highly parallelized method. The paper does not clearly explain how the proposed compression method can manage this parallelization without impacting rendering speed.**
>
> As indicated in lines 529–532 of our paper, **the entropy decoding of the Gaussian primitives and the rendering of images are two distinct and decoupled processes**. The entropy decoding is an one-off operation that decodes the offsets $\\{\hat{O} _ i\\}^K _ {i=1}$, scaling $\hat{l}$, and feature $\hat{f}$ from the compressed bitstream. Only after these parameters have been entropy decoded can the rendering of images be started, as with the existing works `[R1, R2, R3]`. **The rendering of images can be performed multiple times and with any parallelization techniques applicable to ScaffoldGS**.
>
> ---
>
> > **[W2] The paper does not address decoding complexity. While CHARM enhances compression performance, it could significantly impact decoding speed, which in turn may affect rendering speed.**
>
> In Table 2 of our initial submission, we have followed the existing works [R1, R2, R3] to **report the decoding time as a rough indicator of the decoding complexity**. Also, as noted in our previous response [W1], the entropy decoding of Gaussian primitives and the rendering of images are two decoupled processes. CARM is part of the entropy decoding process. Thus, it does **NOT really impact the rendering throughput**.
>
> As per the request, we provide additional results comparing between our schemes with and without CARM (the channel-wise AR model). The results indicate that **CARM has a negligible impact on decoding time**.
>
> [**Note**: In reporting the decoding runtime of our CAT-3DGS in the initial submission, the runtime spent on the rate estimation for coding triplanes was accidentally included, leading to our prolonged decoding time. We have updated our decoding time in Table 2 and provided this ablation experiment in Appendix G of our revised manuscript. Please see the text in red.]
>
> | Scene        | Method               | Triplane Decoding Time (s) | Anchor Attributes Decoding Time (s) | Total Decoding Time (s) |
> |--------------|----------------------|----------------------------|--------------------------------------|-------------------------|
> | **room**     | CAT-3DGS (w/ CARM)   | 11.4                       | 2.2                                  | 13.6                    |
> |              | CAT-3DGS (w/o CARM)  | 11.3                       | 2.1                                  | 13.4                    |
> | **amsterdam**| CAT-3DGS (w/ CARM)   | 47.4                       | 17.0                                 | 64.4                    |
> |              | CAT-3DGS (w/o CARM)  | 47.2                       | 14.9                                 | 62.1                    |
>
> ---
>
> > **[W3] The paper does not clarify whether real entropy coding was used during rendering or if the bit-rate was calculated based on theoretical entropy coding.**
>
> Yes, at inference time, the **real entropy encoding/decoding is performed** to generate compressed bitstreams and report our experimental results. In the revised manuscript, we have updated the text in lines 448-450 accordingly. Please see the text in red color.
>
> ---
>
> > **[Q1] Are there scenarios where CAT-3DGS underperforms? Which modules in CAT-3DGS may contribute to a performance drop?**
>
> A very good question! As shown in Figure 4 of our initial submission, our CAT-3DGS achieves the state-of-the-art compression results across all the test datasets. However, its coding gain over HAC is relatively smaller on the large-scale BungeeNeRF dataset. Currently, the three triplanes are set to have the same spatial dimensions. We conjecture that this setting is sub-optimal for BungeeNeRF, where most of the Gaussian primitives are densely distributed along one major plane.
>
> ---
>
> **We thank the reviewer for their insightful and constructive comments.** We have carefully addressed all the concerns and comments and hope the reviewer find our responses satisfactory. If so, we would greatly appreciated if the reviewer could consider reflecting this in the rating for this early attempt at efficient coding of 3DGS representations, which we believe would invite more contributions from the community to advance this largely unexplored yet important area.
>
> ---
>
> **Reference**\
> `[R1]` X. Liu et al.,"Compgs: Efficient 3d scene representation via compressed gaussian splatting," arXiv, 2024. \
> `[R2]` Y. Chen et al., "Hac: Hash-grid assisted context for 3d gaussian splatting compression," arXiv, 2024.  \
> `[R3]` Y. Wang et al., "Contextgs: Compact 3d gaussian splatting with anchor level context model," arXiv, 2024.

---

> ### Author Response · Authors · 2024-11-25
> **Response to Reviewer egdp**
>
> Dear Reviewer, we thank you very much for the effort you put into reviewing our paper. We have addressed your comments to the extent possible. As the **interactive discussion phase for the rebuttal will end on Nov. 27 AOE**, we look very much forward to your further comments (if any).  If you find our responses satisfactory, we would ask your kind consideration for increasing your rating on this early and novel attempt at addressing 3DGS coding.

---

> > ### Comment · Reviewer_egdp · 2024-11-26
> >
> > Dear Author(s),
> >
> > Thank you for addressing my concerns. I have no further questions.

---

> > > ### Author Response · Authors · 2024-11-27
> > >
> > > Thank you for reviewing our responses carefully. If you find them satisfactory, kindly consider increasing your rating.

---

### Author Response · Authors · 2024-11-20
**Response to All Reviewers**

We thank the reviewers for their insightful and constructive comments. In particular, we appreciate all reviewers for recognizing the effectiveness of our work. We are also pleased that Reviewers (egdp, stVt, and tuJM) found our work **novel**, and that Reviewer (VHE1) considered our **coding architecture to be well-structured and theoretically solid**. Additionally, we are grateful to Reviewers (egdp, tuJM) for appreciating the **comprehensive experimental results**.

We have carefully addressed all the concerns and comments and hope the reviewers find our responses satisfactory. We kindly request that the reviewers consider increasing the ratings for this early attempt at efficient coding of 3DGS representations, which we believe would invite more contributions from the community to advance this largely unexplored yet important area.

---

### Meta-Review · Area_Chair_ExHv · 2024-12-16

**Metareview:**

This is a paper discussing scene compression using a Gaussian splatting model. The paper introduces several novelties (e.g., triplane-based hyperprior, view frequency-aware masking mechanism), and evaluates against the most important baselines. No major weaknesses are noted. The work represents a significant contribution, albeit somewhat incremental.

**Additional Comments On Reviewer Discussion:**

This was a straight-forward review. All reviewers consider the paper a significant contribution, albeit somewhat incremental. Authors were responsive and clarified several aspects of the work.

---

### Decision · Program_Chairs · 2025-01-22

Accept (Poster)